# Fast retrieval and autonomous regulation of single spontaneously recycling synaptic vesicles

Jeremy Leitz[1], Ege T Kavalali[1,2]*

[1]Department of Neuroscience, University of Texas Southwestern Medical Center, Dallas, United States; [2]Department of Physiology, University of Texas Southwestern Medical Center, Dallas, United States

**Abstract** Presynaptic terminals release neurotransmitters spontaneously in a manner that can be regulated by $Ca^{2+}$. However, the mechanisms underlying this regulation are poorly understood because the inherent stochasticity and low probability of spontaneous fusion events has curtailed their visualization at individual release sites. Here, using pH-sensitive optical probes targeted to synaptic vesicles, we visualized single spontaneous fusion events and found that they are retrieved extremely rapidly with faster re-acidification kinetics than their action potential-evoked counterparts. These fusion events were coupled to postsynaptic NMDA receptor-driven $Ca^{2+}$ signals, and at elevated $Ca^{2+}$ concentrations there was an increase in the number of vesicles that would undergo fusion. Furthermore, spontaneous vesicle fusion propensity in a synapse was $Ca^{2+}$-dependent but regulated autonomously: independent of evoked fusion probability at the same synapse. Taken together, these results expand classical quantal analysis to incorporate endocytic and exocytic phases of single fusion events and uncover autonomous regulation of spontaneous fusion.

*For correspondence: ege.kavalali@utsouthwestern.edu

Competing interests: The authors declare that no competing interests exist.

## Introduction

Synaptic terminals release neurotransmitters either spontaneously or in response to presynaptic action potentials (APs) (*Fatt and Katz, 1952*). In addition to the well-established role of AP-evoked neurotransmitter release in information transfer and processing, a growing number of studies assign a key role for spontaneous release in synaptic homeostasis and plasticity (*Sutton and Schuman, 2005*; *Kavalali et al., 2011*; *Hawkins, 2013*). Recent work indicates that these two modes of neurotransmission are largely independent in terms of their presynaptic regulation as well as postsynaptic signaling consequences (*Sara et al., 2005*; *Sutton et al., 2006, 2007*; *Atasoy et al., 2008*; *Melom et al., 2013*; *Nosyreva et al., 2013*; *Wierda and Sorensen, 2014*). However, the molecular mechanisms that underlie the segregation of the two forms of release are only beginning to be elucidated (*Hua et al., 2011*; *Pang et al., 2011*; *Ramirez et al., 2012*; *Bal et al., 2013*; *Zhou et al., 2013*; *Wang et al., 2014*). There is strong evidence that synaptic vesicles recycle at rest in the absence of presynaptic APs and take up exogenous probes such as FM dyes, antibodies or horseradish peroxidase (*Ryan et al., 1997*; *Murthy and Stevens, 1998*; *Sara et al., 2005*; *Peng et al., 2012*; *Kavalali and Jorgensen, 2014*). Studies also suggest that endocytic mechanisms mediating synaptic vesicle retrieval after spontaneous fusion diverge from those that trigger endocytosis after AP-evoked exocytosis (*Chung et al., 2010*; *Peng et al., 2012*; *Meng et al., 2013*). However, visualizing single spontaneous vesicle fusion and retrieval events has been technically difficult as the stochastic nature and low probability of spontaneous fusion requires long-term imaging with high temporal resolution, which typically gives rise to significant photobleaching and potential photodamage. Earlier attempts at detecting spontaneous synaptic vesicle exo-endocytosis using capacitance measurements heavily

**eLife digest** Neurons communicate with one another at junctions called synapses. When an electrical signal known as an action potential arrives at a synapse, it causes packages called vesicles to fuse with the membrane that surrounds the neuron. The vesicles contain molecules called neurotransmitters, which are then released into the gap between the neurons. When these molecules bind to receptors on the surface of the second neuron, a copy of the action potential is generated and travels along the second neuron. The empty vesicles are then reabsorbed back into the first cell to be refilled with neurotransmitters so that the whole process can be repeated.

In addition to releasing neurotransmitters in response to the arrival of an action potential, neurons sometimes release vesicles spontaneously. Such events are relatively rare and occur seemingly at random, making them difficult to study. However, by labeling a synaptic vesicle protein with a fluorescent protein, Leitz and Kavalali have constructed a system in which they can observe spontaneous vesicle fusions in single synapses in cell cultures, and follow the fate of the vesicles as they are reabsorbed back into the cell.

The results reveal a number of key differences between the spontaneous events and those triggered by action potentials. Vesicles released spontaneously are retrieved and recycled much more rapidly than those that are released following the arrival of an action potential. Moreover, increases in calcium levels increase the frequency of both types of events. However, it is also clear that the calcium ions influence the two types of events independently of one another.

Recent research on flies has suggested that some regions of synapses only ever release vesicles spontaneously, whereas others only ever release vesicles in response to the arrival of an action potential. The work of Leitz and Kavalali now adds to increasing evidence that the spontaneous release of neurotransmitters may have its own role in neuronal signaling that is distinct from the role played by neurotransmitters that are released in response to action potentials.

relied on signal averaging and was confounded by susceptibility to contamination by capacitance changes unrelated to synaptic vesicle exocytosis (*Sun et al., 2002*; *Yamashita et al., 2005*).

The current lack of insight into single synaptic vesicle retrieval leaves open the question of whether spontaneous synaptic vesicle exocytosis is tightly and temporally coupled to vesicle retrieval. In this study, we used the vesicular glutamate transporter or the vesicle protein synaptophysin as carriers for luminal pH-sensitive fluorescent probes and optimized imaging conditions to minimize photobleaching without compromising our ability to detect a majority of spontaneous synaptic vesicle fusion events. Our optical recording conditions were similar to our earlier work where we characterized single AP-evoked fusion events with a median probability of 0.2 (*Leitz and Kavalali, 2011*) indicating that these settings enable visualization of release from single boutons (*Murthy et al., 1997*). Under these conditions, we found that single synaptic vesicles are retrieved extremely rapidly (<370 ms) after spontaneous fusion indicating the fluorescence decay of individual events was dominated by vesicle re-acidification. These spontaneous fusion events were coupled to post-synaptic N-methyl-D-aspartate (NMDA) receptor-driven $Ca^{2+}$ signals as supported by their temporal and spatial juxtaposition to D(−)-2-Amino-5-phosphonopentanoic acid (AP-5)-sensitive fluorescence signals originating from a $Ca^{2+}$ indicator targeted to postsynaptic densities. Surprisingly, we uncovered a significant fraction of putative multiquantal events that increased in prevalence at elevated $Ca^{2+}$ concentrations. Furthermore, we could not detect appreciable correlation between the propensities of evoked and spontaneous fusion events at increasing $Ca^{2+}$ concentrations. These experiments demonstrated that spontaneous fusion propensity in a given synapse is regulated autonomously and independently of evoked release probability. Taken together, these results expand classical quantal analysis (*Del Castillo and Katz, 1954*; *Boyd and Martin, 1956*) to incorporate exocytic and endocytic phases of single fusion events and provide insight into the properties and regulation of single spontaneous fusion events in relation to their AP-evoked counterparts that originate from the same synaptic bouton.

## Results

### Synaptic vesicles that fuse spontaneously recycle rapidly

To identify spontaneous fusion events, we first imaged synapses expressing vGlut-pHluorin at 8 Hz to minimize photobleaching and identified small, rapid increases in fluorescence (*Figure 1A–D*) in the presence of the voltage-gated Na$^+$ channel blocker, tetrodotoxin (TTX). These increases in fluorescence were distinguishable from noise and could be fit with a single Gaussian curve of mean amplitude of 623 ± 122 a.u (*Figure 1B,C*) and average ΔF/F of 3.8 ± 0.93%. These events decayed rapidly with a weighted average decay time constant (heretofore referred to as 'decay time') of 0.29 ± 0.017 s. However, the decay times of these events were distributed according to a non-geometric β-function with a non-geometric average fluorescence decay time of 0.37 s and median decay time of 0.28 s (*Figure 1D*). Moreover, in contrast to our earlier observations (*Leitz and Kavalali, 2011*), the fluorescence decay of these spontaneous events proceeded immediately following fluorescence increase without any observable dwell time. To confirm that these rapidly decaying increases in fluorescence were due to genuine spontaneous vesicle fusion, we sought to manipulate the rate of decay by addition of 50 mM Tris–HCl in the extracellular solution. Indeed, equiosmolar substitution of extracellular NaCl with 50 mM Tris–HCl slowed vesicle reacidification to an average decay time of 0.55 s with median decay time 0.32 s (*Figure 1E,F*) indicating that the decay phase of these events was dominated by vesicle re-acidification. We then compared these spontaneous fluorescence increases to those evoked by single action potential stimulation in the absence of TTX (*Figure 1G–I*). We found that fusion events evoked by stimulation had a mean fluorescence amplitude of 641 ± 171 a.u. (*Figure 1H*), similar to spontaneous fluorescence increases. However, these events decayed more slowly with average decay time of 0.83 s and median decay time of 0.83 s (*Figure 1I*). Finally, to evaluate whether we were able to visualize the entirety of the fluorescent signal originating from spontaneous vesicle fusion events, we incubated neurons with the vacuolar ATPase inhibitor, folimycin (80 nM), and measured spontaneous increases in fluorescence (*Figure 1J,K*). In the presence of folimycin, increases in fluorescence take on a staircase-like waveform with mean amplitude of 609 ± 195 a.u. (*Figure 1K*), similar to that of spontaneous fusion events observed in the absence of folimycin. There were no significant differences in the mean amplitude between spontaneous increases in fluorescence with or without folimycin and those that were evoked by stimulation (ANOVA p value >0.5). However, when comparing amplitude distributions there was a significant difference between spontaneous events in the absence of folimycin and both spontaneous events in the presence of folimycin and action-potential evoked single vesicle fusion events (KS$_{test}$ < 0.05 Dmax = 0.08 at bin 450 a.u.) (*Figure 1L*). This result suggests that while we were able to acquire the entire fluorescence waveform of the majority of spontaneous fusion events, there remains a small population of low amplitude events that we were unable to detect due to the rapid decay of fluorescence signals. Regardless, our results still indicate a clear divergence in the reacidification rate between vesicles that recycle at rest and those that fuse in response to action potentials.

Finally, to assess the contribution of false positives to our analysis, we employed two approaches. First, we used the same detection criteria to identify potential 'hits' with a negative amplitude (white bars in *Figure 1D*), that is, 'events' that were less than −2 times the standard deviation of the points prior in 2 mM Ca$^{2+}$. Setting our detection criteria for negative amplitudes revealed a false 'hit' rate of 0.065 per bouton per minute, suggesting that false hits contribute a negligible component to our measurements. Second, we analyzed the positive and negative amplitudes of transient fluorescence fluctuations seen in 0 mM Ca$^{2+}$ where genuine event frequency as verified by the presence of folimycin (leading to staircase increases in fluorescence) is extremely low (29 events in 10 min from 200 puncta from four coverslips, that is ~0.01 events/bouton/min compared to ~0.5 events/bouton/min in 2 mM Ca$^{2+}$). Despite the presence of folimycin these false positive events decayed back to baseline and had a significantly lower fluorescence amplitude distribution (KS$_{test}$ < 0.05; *Figure 1L*). When these 0 mM Ca$^{2+}$ experiments were analyzed for a negative amplitude, the absolute value of these negative false events were not significantly different from the positive amplitude false events (KS$_{test}$ > 0.3; *Figure 1L*). Taken together these data suggest that our detection criteria reliably identify events due to genuine spontaneous vesicle fusion with a low level of false positive hits (~10% per recording).

### Dual color imaging shows that spontaneously fusing vesicles elicit postsynaptic Ca$^{2+}$ signals

To investigate the relationship between these fast spontaneous increases in presynaptic fluorescence to postsynaptic receptor activation, we moved to a dual-color system utilizing the red-shifted pHluorin

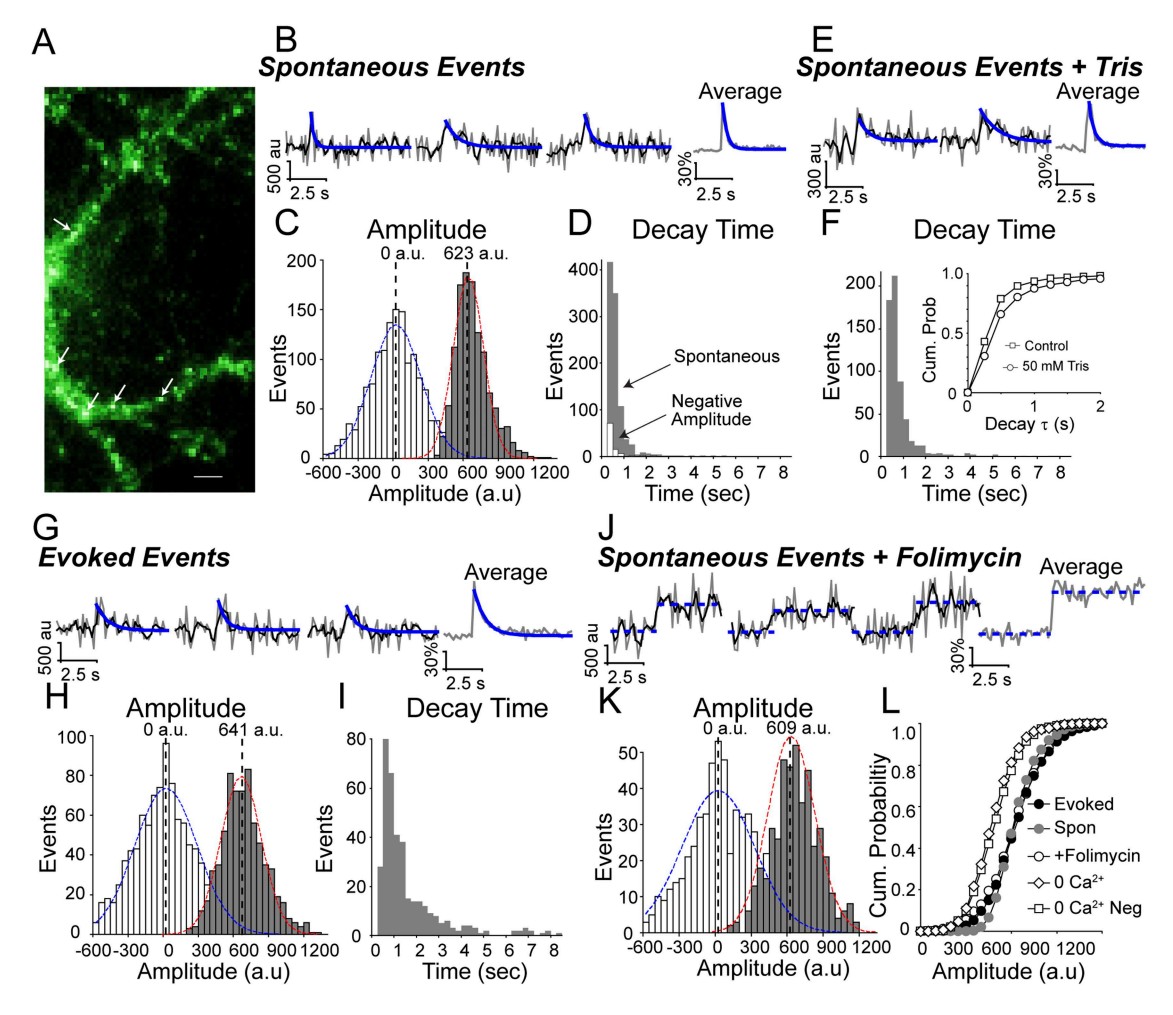

**Figure 1**. Spontaneous increases in vGlut1-pHluorin fluorescence decay rapidly. vGlut1-pHluorin was expressed via lentiviral infection in dissociated hippocampal cultures and neurons were imaged at 16–19 days in vitro. (**A**) Example image of vGlut1-pHluorin expression in NH$_4$Cl (20 mM). Arrows indicate putative synapses. Scale bar is 5 µm. (**B**) Images were recorded in the presence of TTX, AP-5 and CNQX and spontaneous increases in fluorescence were observed. Example traces of spontaneous increases in fluorescence (3 left traces) and the average of all events in this experiment (right trace) are shown. Raw data is in grey, black trace is the moving average of three points, and the blue trace is a fit to a first order decay. (**C**) The amplitudes of spontaneous increases in fluorescence are distinguishable from noise. White bars are the amplitude change of a random section of the fluorescence recording, while successful events are in grey (for detection criteria see 'Materials and methods'). Blue dashed line is a Gaussian fit centered at 0 a.u. and standard deviation of 212 a.u. ($\chi^2 = 0.9$). Red dashed line is a Gaussian distribution fit to the data with mean of 623 ± 122 a.u. and CV = 0.17 (n = 1178 events from four coverslips prepared from two cultures). (**D**) Distribution of decay times of spontaneous increases in fluorescence (grey) have a non-geometric (Beta-function) average of 0.37 s with upper bound = 7 and lower bound = 2 (n = 1178 events from four coverslips prepared from two cultures). Negative amplitude events from the same puncta (white) have a non-geometric (Beta-function) average of 0.20 s (n = 98 events from the same four coverslips). (**E**) Example traces of spontaneous increases in fluorescence in the presence of high (50 mM) Tris-buffered extracellular solution. (**F**) The resulting spontaneous increases in fluorescence were slower to decay with average decay time of 0.55 s with upper bound = 3.8 and lower bound = 2.1 (n = 1212 events from three coverslips generated from two cultures.) Inset shows cumulative probability distribution of decay time in cells with extracellular solution containing HEPES (Control; squares) and 50 mM Tris (circles) (KS$_{test}$ p < 0.001 with Dmax = 0.13 at 0.5 s). (**G**) Example traces of increases in fluorescence in response to single action-potential stimulation delivered at 0.05 Hz. (**H**) Amplitudes of stimulated increases in fluorescence are distinguishable from noise (white bars and blue Gaussian distribution) and could be fit with a Gaussian distribution centered at 641 ± 117 a.u. and CV = 0.26 (red dashed line; $\chi^2 = 0.94$). (**I**) Increases in fluorescence due to single-action potential stimulation delivered at 0.05 Hz were slower to decay with a non-geometric average of 0.83 s with upper bound = 4.4 and lower bound 3.6 and median decay time of 0.83 s (n = 694 events from four coverslips generated from two cultures). (**J**) Example traces of spontaneous increases in fluorescence in the presence of folimycin (80 nM). With inhibition of the vATPase, vesicles cannot be reacidified and therefore increases in fluorescence do not decay (n = 445 events from four coverslips generated from two cultures). (**K**) Amplitude distribution of spontaneous increases in fluorescence in the presence of folimycin are also distinguishable from noise and can be fit with a Gaussian curve with mean amplitude of 609 ± 195 a.u. and CV = 0.32 ($\chi^2 = 0.87$). (**L**) Cumulative probability histogram of amplitudes of spontaneous increases in

*Figure 1. Continued on next page*

*Figure 1. Continued*

fluorescence in 2 mM extracellular $Ca^{2+}$ in the absence (open square) and presence of folimycin (grey square), and amplitudes of fluorescence increases evoked by stimulation (black square). The amplitude distributions of false positive events in 0 mM extracellular $Ca^{2+}$ and folimycin (open diamond) and negative amplitude distributions of 0 mM extracellular $Ca^{2+}$ false positive events (open triangle) were significantly smaller than amplitudes of putative spontaneous events with or without folimycin ($KS_{test}$ p < 0.05 for both false positive and negative amplitude) or stimulation-evoked fusion events ($KS_{test}$ p < 0.05 again for both conditions), but were not significantly different from one another ($KS_{test}$ p > 0.25 between false positives and negative amplitude distributions). The only significant difference between all distributions was between spontaneous increases in fluorescence with and without folimycin ($KS_{test}$ p < 0.05 Dmax = 0.08 at bin 450 a.u.).

variant, pHTomato, fused to the presynaptic vesicle protein synaptophysin (SypHTomato) (*Li and Tsien, 2012*) and the green fluorescent $Ca^{2+}$ indicator, GCaMP5K, fused to the post-synaptic density protein 95 (PSD-95-GCaMP5K) (*Akerboom et al., 2012*). Additionally, we excluded extracellular $Mg^{2+}$ to allow $Ca^{2+}$ influx through postsynaptic NMDA receptors. In this setting we can verify presynaptic vesicle fusion events in the red channel using the resulting $Ca^{2+}$ influx in the green channel as a coincidence detector of a successful presynaptic fusion event (*Figure 2*). However, this system is not without caveats: first, because we are now monitoring two wavelengths our temporal resolution decreased from >8 Hz to ~4 Hz; second, SypHTomato results in an elevated surface expression level compared to vGlut-pHluorin that required a post-hoc decay correction to compensate for photobleaching (See 'Materials and methods'). Despite these issues, spontaneous increases in SypHTomato fluorescence were still detected and were distinguishable from noise with mean amplitude of 303 ± 92 a.u. (*Figure 2C*) and ΔF/F of 1.9 ± 0.67%. In 2 mM extracellular $Ca^{2+}$ approximately 69% of spontaneous increases in SypHTomato elicited a PSD-95-GCaMP5K signal within ±1 frame, increasing to 83% between −1 to +5 frames. We include this −1 group because in our system we obtain only one time point for each cycle of imaging (i.e. one frame of red channel and one frame of green channel are both assigned the same time point). Therefore, fusion events that may occur halfway through an imaging cycle would appear shifted by −1 frame. Spontaneous increases in fluorescence that were correlated (within ±1 frame) with $Ca^{2+}$ signals were distinguishable from noise with mean amplitude of 170 ± 50 a.u. (*Figure 2D*) and ΔF/F of 2.2 ± 1.3%. We next compared the amplitudes of these spontaneous fusion events with increases in fluorescence elicited by single action potential stimulation (*Figure 2E,F*). Stimulation resulted in events that were distinguishable from noise with mean amplitude of 345 ± 95 a.u. (*Figure 2F*). Although both spontaneous and AP-evoked fluorescence increases were distinguishable from noise, the small amplitude of these events combined with the rapid decay times made precise measurements of individual decay times difficult. Therefore we averaged all events within an experiment to estimate the overall average decay times (*Figure 2G*). We found that, as with vGlut-pHluorin, spontaneous fluorescence increases in SypHTomato decayed faster than those evoked by stimulation with average decay times of 0.5 ± 0.2 s and 1.0 ± 1.5 s, respectively. To confirm that we were able to observe the full fluorescence increase of spontaneous fusion events with our reduced temporal resolution, we incubated neurons in folimycin and observed stepwise increases in SypHTomato fluorescence. In the same experiments to verify that the increases in the GCaMP5K fluorescence were due to $Ca^{2+}$ influx through NMDA receptors, we perfused an extracellular solution containing the NMDA receptor blocker AP-5 (*Figure 2H–S*). We were able to identify spontaneous fusion events according to stepwise increases in SypHTomato fluorescence alone without relying on the coincidence with GCaMP5K mediated $Ca^{2+}$ signals (*Figure 2H–S*). Under these conditions, we observed spontaneous fusion events with mean amplitude of 316 ± 64.9 a.u (*Figure 2L*) similar to both those in the absence of folimycin and those due to stimulation. Events in the same synapses in the presence of AP-5 were also similar with mean amplitude of 270 ± 65 a.u. (*Figure 2M*). The GCaMP5K signal in the absence of AP-5 had mean amplitude of 153 ± 57.6 a.u. (*Figure 2R*), similar to without folimycin, but in the presence of AP-5, the GCaMP signal amplitude decreased to the baseline noise, 13.2 ± 56.0 a.u. (*Figure 2S*). Together these data complement our observations using vGlut1-pHluorin and show that spontaneous vesicle fusion events are coupled to postsynaptic NMDA receptor-driven $Ca^{2+}$ signals and, following exocytosis, these vesicles are retrieved and re-acidified on a much faster time course than their AP-evoked counterparts.

## Increasing extracellular $Ca^{2+}$ triggers multiple successive spontaneous fusion events

Using presynaptic imaging similar to that used in this study, we have previously shown that increasing extracellular $Ca^{2+}$ increased synaptic vesicle fusion probability and increased the likelihood of

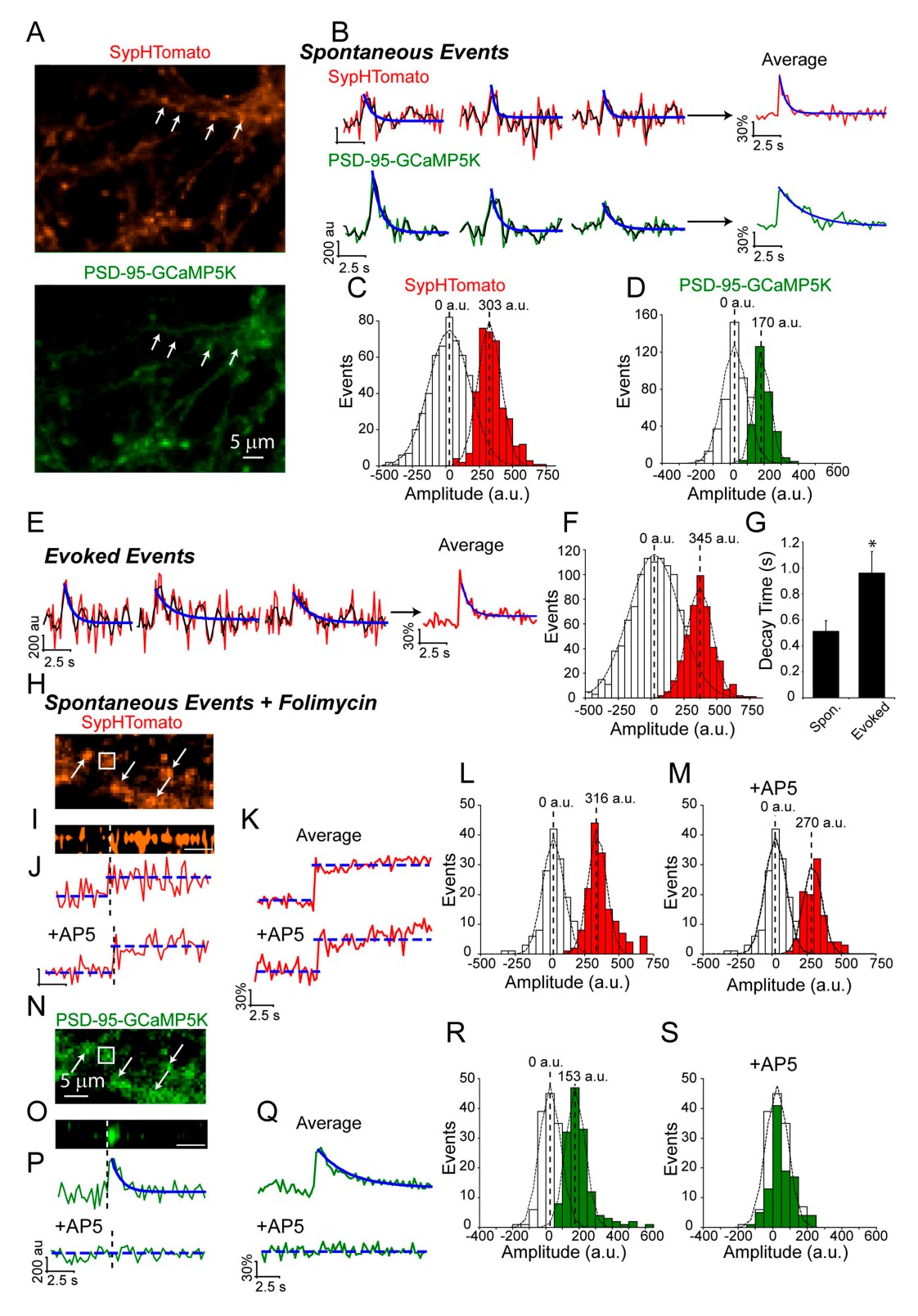

**Figure 2**. Dual color imaging shows that spontaneous fusion events are coupled to postsynaptic NMDA receptor-driven Ca²⁺ signals. (**A**) Example images of SypHTomato and PSD-95-GCaMP5K expression. Arrows indicate putative synapses. Scale bar is 5 μm. (**B**) Example traces of SypHTomato (raw data in red, a moving average of three points in black and a fit of the decay time t in blue) and PSD-95-GCaMP5K (raw data in green, a moving average of three

*Figure 2. Continued on next page*

*Figure 2. Continued*

points in black and a fit of the decay time in blue). Because spontaneous increases in fluorescence were very small, we averaged events for each experiment representative average traces are shown at right. (**C**) The amplitude distribution of SypHTomato could be well fit with a Gaussian curve centered at 303 ± 92 a.u. ($\chi^2$ = 0.86) (n = 361 events from four coverslips generated from two cultures). (**D**) Amplitudes of PSD-95-GCaMP were distinguishable from noise and could be fit with a Gaussian curve with mean amplitude of 170 ± 50 ($\chi^2$ = 0.98) (n = 361 events from four coverslips generated from two cultures). (**E**) Example traces of SypHTomato fluorescence in response to single action potentials delivered at 0.05 Hz. Again, raw data is in red, a moving average of 3 points is in black and the decay time fit is in blue (n = 443 events from four coverslips generated from two cultures). (**F**) Amplitudes of fluorescence increases evoked by action-potential stimulation could be well fit by a Gaussian curve with mean of 345 ± 95 a.u ($\chi^2$ = 0.99). (**G**) Averaged traces of spontaneous increases in fluorescence decayed back to baseline with decay time = 0.51 ± 0.08 s (n = 4), while events that responded to stimulation (n = 3) decayed much slower t = 0.96 ± 0.06 (Student's t-test p value <0.05). (**H**) Example image of SypHTomato expression in the presence of folimycin. Arrows indicate putative synapses. The box is the region from which a line scan was taken (shown in panel I). (**I**) Line scan of SypHTomato fluorescence. White dashed line indicates where on the corresponding trace the fluorescence step occurred, scale bar = 2.6 s. (**J**) Example traces of events in the presence and absence of AP-5 from the same synapses. (**K**) Average of traces from the experiment of step-wise increase in fluorescence in the presence and absence of AP-5. (**L**) Increases in fluorescence in the presence of folimycin were separable from noise and could be fit with a Gaussian with mean amplitude of 316 ± 65 a.u. ($\chi^2$ = 0.49) (n = 154 events from four coverslips generated from two cultures). (**M**) The same synapses in the presence of AP-5 showed spontaneous increases in fluorescence that were still distinguishable from noise albeit with a slightly smaller amplitude distribution 270 ± 65 a.u. ($\chi^2$ = 0.48) that was not significantly different from amplitudes in the presence of folimcyin (not shown; $KS_{test}$ p < 0.01) (n = 103 events from four coverslips generated from two cultures). (**N**) Example image of corresponding PSD-95-GCaMP5K fluorescence. White arrows indicate putative synapses. The box is the region from which a line scan was taken. (**O**) Line scan of PSD-95-GCaMP5K fluorescence signal that occurred at the same time as the above SypHTomato signal. (**P**) Example traces of events in the presence and absence of AP-5. (**Q**) Average of traces in the presence and absence of AP-5. In the presence of AP-5, entry of $Ca^{2+}$ into the postsynaptic terminal is prevented and thus there is no GCaMP5K signal. (**R**) PSD-95-GCaMP5K signals were separable from noise and could be fit with a Gaussian curve with mean amplitude of 153 ± 58 a.u. ($\chi^2$ = 0.80) (n = 154 events from four coverslips generated from two cultures). (**S**) In the presence of AP-5, $Ca^{2+}$ is prevented from entering the postsynaptic terminal and results in no detectable GCaMP5K events (n = 103 events from four coverslips generated from two cultures.).

multivesicular fusion events, which is observed as an increase in fluorescence amplitude (*Leitz and Kavalali, 2011*). To assess if this scenario also applied to vesicles that fuse spontaneously, we measured the amplitude of spontaneous increases in vGlut-pHluorin fluorescence in 2, 4 and 8 mM extracellular $Ca^{2+}$ (*Figure 3A–C*). Surprisingly, we found that fluorescence amplitude distributions even in 2 mM extracellular $Ca^{2+}$ did not fit well to a single Gaussian curve (D'Agostino-Pearson omnibus K2 normality test p < 0.02), nor could the sum of two Gaussian curves with means at integer multiples (i.e. a quantal distribution of amplitudes indicative of two or more vesicles undergoing fusion simultaneously) account for the distribution of amplitudes (*Figure 1B* red lines). Instead, amplitude distributions in 2 and 4 mM extracellular $Ca^{2+}$ (*Figure 3D,E*, respectively) were best fit by the sum of two Gaussian curves at 1 quantal mean (1q = 623 ± 91 a.u., a slightly smaller standard deviation than in Figure 1) and 1.3 times the first mean (1.3q = 834 ± 111 a.u.) (*Figure 3D,E* black lines). Fluorescence amplitudes in 8 mM $Ca^{2+}$ were best fit by the sum of three Gaussian curves distributed at 1 quantal mean (1q = 623 ± 91 a.u.), 1.3 times the first quantal mean (1.3q = 834 ± 111 a.u.) and 2 times the first quantal mean (1246 ± 182 a.u.) (*Figure 3F*). This shift in fluorescence may arise from the limitations of our sampling rate and the rapid fluorescence decay time of these spontaneous events, therefore the 1.3q events may reflect two spontaneous events occurring close together in time but with a slight delay such that at our acquisition rate of >8 Hz, they appear to be a single event with a normalized amplitude of 1.3q (*Figure 3G*). To estimate the delay between two events that would be required for such a fluorescence signal, we modeled the fluorescence decay of two hypothetical spontaneous fusion events with randomly generated amplitudes within the observed first Gaussian distribution (1q) and fixed decay times of 371 ms (the average decay time of spontaneous events in 2 mM $Ca^{2+}$). We used this model to generate 2000 hypothetical decay times and found that a fusion delay of 118 ms would result in an amplitude ~1.3 times greater than the single quantal amplitude. We then estimated the percent of fusion events that were multivesicular (*Figure 3H*) and found that at 2 mM $Ca^{2+}$ 17% of events were multivesicular, increasing to ~30% in 4 mM $Ca^{2+}$ and up to 70% of events were multivesicular in 8 mM $Ca^{2+}$. While this analysis favors preponderance of multivesicular spontaneous release at elevated $Ca^{2+}$ concentrations, this shift in fluorescence amplitudes could be also due to unequal distribution of pHluorin molecules within the population of spontaneously recycling synaptic vesicles selectively mobilized at higher $Ca^{2+}$ concentrations. In addition, this fluorescence shift could arise from the preferential fusion, at elevated $Ca^{2+}$ conditions, of larger synaptic vesicles that contain more fluorophores. Given their cell biological complexity (i.e. non-homogenous impact of elevated $Ca^{2+}$

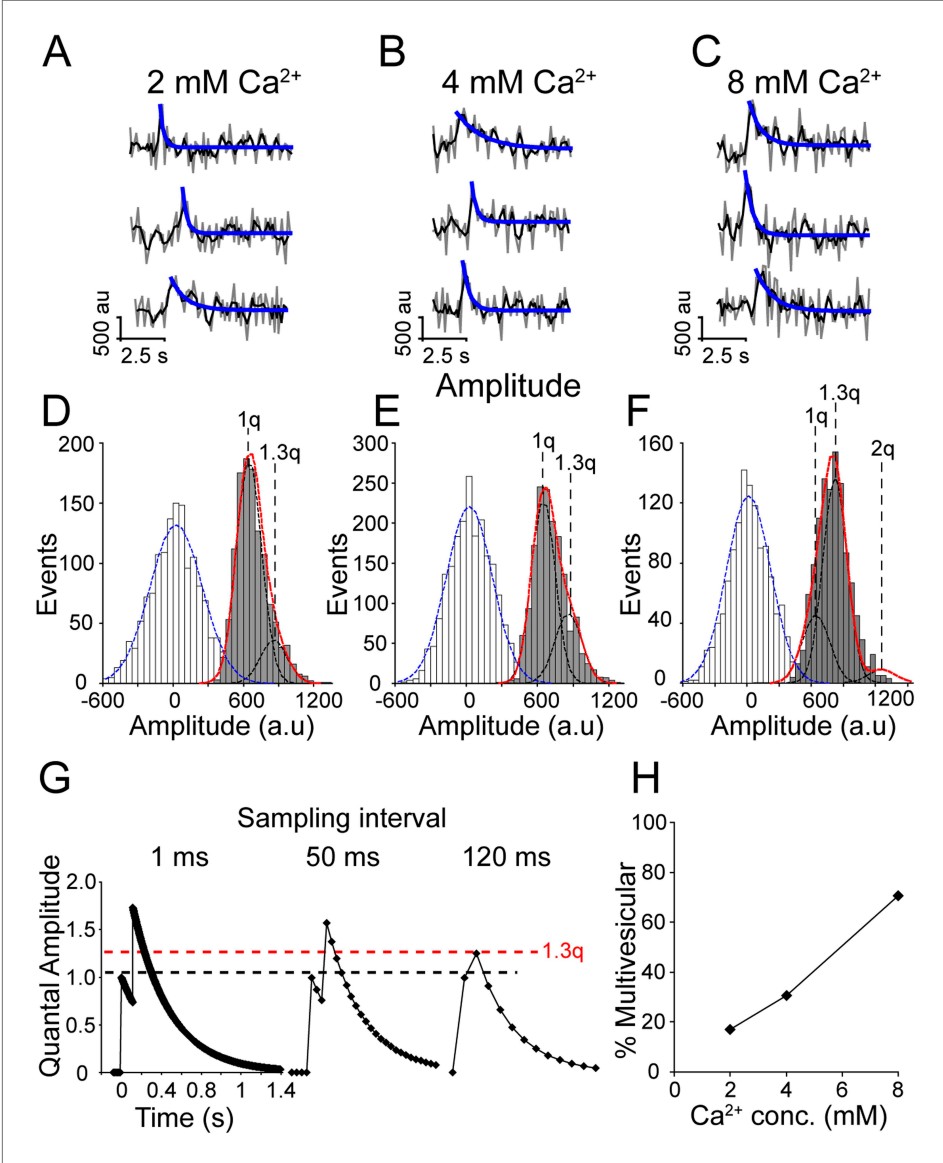

**Figure 3**. Increasing extracellular Ca2+ increases multivesicular release. (**A–C**) Example traces of events in 2 mM (**A**), 4 mM (**B**) and 8 mM (**C**) extracellular Ca2+. Raw data are shown in grey, a moving average of 3 points is shown in black and the blue line indicates a fit of the decay. (**D–F**) Fluorescence amplitude distributions. White bars are noise fit by blue Gaussian distribution centered at 0 a.u., grey bars are successful fusion events fit by multiple Gaussian curves (black lines) and a sum of Gaussian curves (in red). (**D**) Fluorescence amplitudes in 2 mM extracellular Ca2+ were best fit by the sum of two Gaussian curves with mean amplitudes 623 ± 122 a.u. (1q) and 834 ± 111 a.u. (1.3q) ($\chi^2$ = 0.88). (n = 1178 events from four coverslips generated from two cultures). (**E**) Fluorescence amplitudes in 4 mM extracellular Ca2+ were well fit with two similar Gaussian curves 623 ± 98 a.u. (1q) and 834 ± 111 a.u. (1.3q) ($\chi^2$ = 0.70) (n = 1593 events from six coverslips generated from two cultures). (**F**) Fluorescence amplitudes of 8 mM Ca2+ were best fit with the sum of three Gaussian curves with mean amplitudes 623 ± 91 a.u. (1q), 834 ± 111 a.u. (1.3q) and 1246 ± 182 a.u. (2q) ($\chi^2$ = 1) (n = 1110 events from three coverslips generated from two cultures). (**G**) Model of two hypothetical events with normalized amplitudes of 1, decay times of 371 ms, and delay of onset of 118 ms, the same two events sampled at 120 ms results in fluorescence amplitude of 1.3 times a single event (1.3q). (**H**) The percent of events in 2, 4 and 8 mM extracellular Ca2+ that have amplitudes greater than a single Gaussian distribution were defined as multivesicular. This suggests that there are more multivesicular events at increasing extracellular Ca2+ concentrations.

The following figure supplement is available for figure 3:

**Figure supplement 1**. Increasing extracellular Ca2+ increases amplitude of spontaneous events detected by SypHTomato and PSD-95-GCaMP5K.

concentrations on vesicle populations with systematic differences) we consider these two alternative scenarios unlikely. Finally, it is important to note that although this observed increase in event amplitude could be due to vesicle fusion from multiple synapses within our region of interest. This is more likely at elevated $Ca^{2+}$ concentrations as the release probability estimates we present here at 2 mM $Ca^{2+}$ are consistent with release from a single release site (*Murthy et al., 1997*).

We also detected a similar $Ca^{2+}$-dependent increase in the amplitude of spontaneous events identified in SypHTomato/PSD-95-GCaMP5K expressing neurons (*Figure 3—figure supplement 1A–F*). In these neurons, there was an increase in GCaMP5K event amplitudes as a function of increasing extracellular $Ca^{2+}$ indicative of the increased driving force of $Ca^{2+}$ influx. Taken together, these data suggest that increasing extracellular $Ca^{2+}$ increases the probability of two spontaneous fusion events occurring in close temporal proximity within a single synapse (or possibly in adjacent release sites).

## $Ca^{2+}$ does not alter the kinetics of fluorescence transients originating from spontaneous fusion events

We have previously shown that increasing extracellular $Ca^{2+}$ increases the fluorescence decay times of vGlut1-pHluorin containing vesicles that fuse in response to stimulation (*Leitz and Kavalali, 2011*). Here, we wanted to determine if this property was applicable to vesicles that undergo fusion in the absence of stimulation. We increased extracellular $Ca^{2+}$ to 4 mM and found that fluorescence signals decay with an average decay time of 0.44 s (with upper bound = 4.8 and lower bound = 2.1) and median decay time of 0.29 s (*Figure 4A*). In 8 mM extracellular $Ca^{2+}$, the average decay time was identical (0.44 s with upper bound = 4.8 and lower bound = 2.1), and the median decay time did not appreciably change from 0.29 s to 0.28 s (*Figure 4B*). When compared to 2 mM extracellular $Ca^{2+}$ there was no significant difference in either 4 or 8 mM extracellular $Ca^{2+}$. Compared to the decay times of evoked-fusion events, decay times of spontaneous fusion events—regardless of extracellular $Ca^{2+}$ concentration—were much faster (*Figure 4C*). These data suggest that there is a fundamental difference in the kinetics of endocytosis and reacidification between vesicles that fuse spontaneously and those that fuse in response to stimulation.

Our earlier work showed that an increase in the number of vesicles that fuse slows the fluorescence decay time of fusion events (*Leitz and Kavalali, 2011*). Therefore, here, we analyzed the decay times of events in 8 mM $Ca^{2+}$ as a function of amplitude and found that there was no correlation in event size and fluorescence decay times (*Figure 4D*). Furthermore, we also found a similar trend of $Ca^{2+}$-independence in neurons expressing SypHTomato/PSD-95-GCaMP5K (*Figure 4—figure supplement 1A and B*). All of these decay times, regardless of extracellular $Ca^{2+}$ levels, were faster than those observed during stimulation-evoked fusion (*Figure 4—figure supplement 1C*). It is important to note that the rate of decay is approaching the limit of our temporal resolution, which is lowered in an attempt to reduce photobleaching during long imaging episodes required to identify spontaneous fusion events. Thus, it is possible that we either cannot detect a significant change in decay times, or we are missing a subset of ultra fast decay times. Regardless, these data not only suggest that the mechanisms controlling the rate of synaptic vesicle reacidification are not the same for spontaneous and stimulation-evoked vesicle fusion but that endocytosis of synaptic vesicles released at rest is rapid even during multivesicular fusion events, unlike retrieval of vesicles released in response to stimulation.

## Increasing extracellular $Ca^{2+}$ concentration increases the frequency of fast spontaneous vesicle fusion and retrieval

It is well established that increasing the concentration of extracellular $Ca^{2+}$ increases spontaneous vesicle fusion rate at rest (*Lou et al., 2005*; *Sun et al., 2007*; *Xu et al., 2009*). Next, we wanted to know if we see the same increase in our system. Surprisingly, we found no significant change in vesicle fusion frequency between 2 mM and 8 mM extracellular $Ca^{2+}$ (with 0.56 ± 0.02 fusion events per min in 2 mM extracellular $Ca^{2+}$ and 0.49 ± 0.02 fusion events per min in 8 mM extracellular $Ca^{2+}$; p value >0.1) beyond the increase in multivesicular events we reported earlier (*Leitz and Kavalali, 2011*) (*Figure 5A*). We then counted all events with amplitudes within the first quantal mean as a single event, and events with larger amplitudes as two events (*Figure 5B*). Even with this criterion we did not detect a significant difference between 2 mM extracellular $Ca^{2+}$ and 8 mM extracellular $Ca^{2+}$ (with 0.70 ± 0.02 and 0.76 ± 0.03 fusion events per min in 2 and 8 mM extracellular $Ca^{2+}$, respectively; p value >0.1). However, when vesicle reacidification was buffered using 50 mM Tris–HCl (*Figure 5C*)

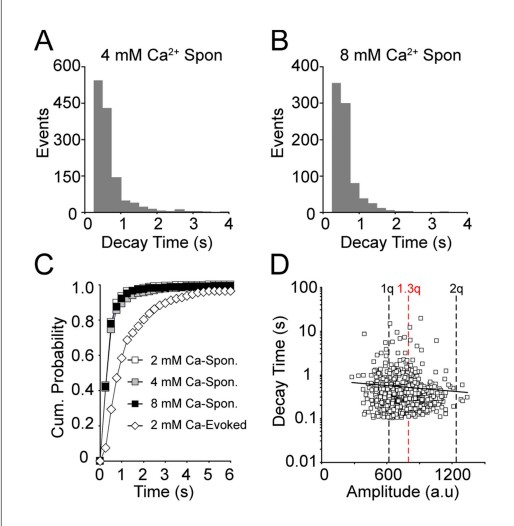

**Figure 4**. Increasing extracellular $Ca^{2+}$ does not alter fluorescence decay time. (**A**) Distribution of fluorescence decay times in 4 mM extracellular $Ca^{2+}$ can be fit with a beta-distribution with mean of 0.44 s with upper bound = 4.8 and lower bound = 2.1 ($\chi^2$ = 0.9; n = 1308 events from six coverslips over three cultures). (**B**) Distribution of fluorescence decay times in 8 mM extracellular $Ca^{2+}$ were fit with a beta-distribution with mean of 0.44 s with upper bound = 4.8 and lower bound = 2.1 ($\chi^2$ = 1; n = 844 events from three coverslips from two cultures). (**C**) Cumulative probability histogram of decay times of spontaneous fluorescence events in 2, 4, and 8 mM extracellular $Ca^{2+}$ showed no significant difference in decay time distributions (e.g. $KS_{test}$ 2 mM $Ca^{2+}$ vs 8 mM $Ca^{2+}$: p > 0.3 with Dmax = 0.04 at 0.5 s). However, the decay times of all spontaneous events were significantly different from the decay times of evoked-fusion events (for all comparisons $KS_{test}$ p < 0.01 with Dmax = 0.5 s). (**D**) Decay time did not correlate with amplitude of spontaneous events in 8 mM $Ca^{2+}$ ($R^2$ = 0.0014). 1q is the Gaussian mean of one event while 2q is the Gaussian mean of two simultaneous events, 1.3q is the amplitude calculated in *Figure 3*.

The following figure supplement is available for figure 4:

**Figure supplement 1**. Increasing extracellular Ca2+ does not alter decay time of spontaneous increases in sypHTomato fluorescence.

we were able to detect a slight shift in vesicle fusion rate in 8 mM extracellular $Ca^{2+}$ compared to 2 mM $Ca^{2+}$ (0.42 ± 0.02 fusion events per min in 2 mM $Ca^{2+}$ and 0.53 ± 0.02 events per min in 8 mM extracellular $Ca^{2+}$; p value <0.05). Addition of folimycin further exacerbated this effect and clearly showed a ~2.7-fold increase in spontaneous fusion rate as a function of extracellular $Ca^{2+}$ (*Figure 5D*; 0.41 ± 0.02 events per min in 2 mM extracellular $Ca^{2+}$ and 1.1 ± 0.02 events per min in 8 mM extracellular $Ca^{2+}$; p value <0.05). This increase in spontaneous fusion rate comes close to the threefold to fourfold increase we detect using postsynaptic electrophysiological recordings under the same conditions (data not shown). Note that in both the Tris-buffered and folimycin-containing experiments, events were analyzed irrespective of amplitude. Taken together, these data indicate that increasing extracellular $Ca^{2+}$ increases the vesicle fusion rate in our system in a manner that is detectable in the presence of folimycin. However, this increase in spontaneous fusion rate cannot be detected without increasing extracellular pH buffering or inhibiting vesicle re-acidification, indicating that elevated extracellular $Ca^{2+}$ specifically increases fusion of synaptic vesicles that are retrieved and re-acidified rapidly, below the temporal resolution of our imaging protocol. This finding suggests that although elevated $Ca^{2+}$ levels do not alter the kinetics of slower events detectable without the aid of altered re-acidification, $Ca^{2+}$ elevation generates a new population of events that are faster in their kinetics. This observation is consistent with earlier work, which demonstrated that increasing extracellular $Ca^{2+}$ facilitates the propensity of vesicle retrieval events with fast kinetics (*Ales et al., 1999*; *Wu et al., 2009*).

## The rate of spontaneous synaptic vesicle fusion and the probability of stimulation-evoked fusion do not correlate within a given synapse

Next, to evaluate the relationship between the rate of spontaneous synaptic vesicle fusion and the probability of evoked vesicle fusion in a synapse, we incubated neurons in folimycin in order to visualize all events and delivered single action

potential stimulations with long inter-stimulus intervals (15–30 s). Increases in fluorescence that occurred within 1 s of a stimulation were considered to be due to evoked release (note that we cannot then differentiate between synchronous vesicle fusion and fast asynchronous fusion) while increases in fluorescence that did not match with a stimulation time were labeled as spontaneous vesicle fusion events (*Figure 6A*). We found that while increasing extracellular $Ca^{2+}$ concentrations increased both the overall rate of spontaneous fusion (from 0.52 ± 0.028 events per min at 2 mM $Ca^{2+}$ to 0.76 ± 0.049 events per min in 8 mM extracellular $Ca^{2+}$) as well as the overall mean evoked fusion probability (from 0.15 ± 0.008 in 2 mM $Ca^{2+}$ to 0.44 ± 0.016 in 8 mM extracellular $Ca^{2+}$), at increasing $Ca^{2+}$ concentrations the spontaneous vesicle fusion rate and evoked fusion probability estimates obtained from individual

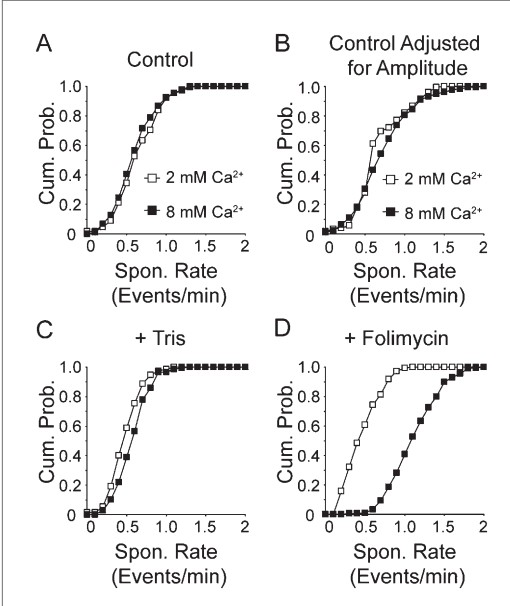

**Figure 5**. Increasing extracellular $Ca^{2+}$ increases spontaneous fusion rate. (**A**) Cumulative probability histogram of spontaneous event rate per synapse per minute in 2 mM and 8 mM extracellular $Ca^{2+}$ (n = 175 synapses from four coverslips for both conditions, $KS_{test}$ p = 0.55 with Dmax = 0.8 fusion events per minute per synapse, with averages of 0.56 ± 0.02 and 0.49 ± 0.02 (Student's t-test p value = 0.2) events per minute per synapse for 2 and 8 mM $Ca^{2+}$, respectively). (**B**) Cumulative probability histogram of spontaneous event rate per synapse per minute in 2 mM and 8 mM extracellular $Ca^{2+}$ adjusted to count large amplitude events as two vesicle fusion events (n = 175 synapses from four coverslips for both conditions, $KS_{test}$ p < 0.05 with Dmax = 0.6 fusion events per minute per synapse, with averages of 0.70 ± 0.02 and 0.76 ± 0.03 (t-test p value = 0.1) events per minute per synapse for 2 and 8 mM $Ca^{2+}$, respectively. (**C**) Cumulative probability histogram of spontaneous event rate per synapse per minute in Tris-buffered (50 mM) 2 mM and 8 mM extracellular $Ca^{2+}$ solutions (n = 150 synapses from three coverslips for both conditions, $KS_{test}$ p < 0.05 with Dmax = 0.5 fusion events per minute per synapse, with average 0.4 ± 0.02 and 0.5 ± 0.02 (t-test p value <0.05) fusion events per minute per synapse for 2 and 8 mM $Ca^{2+}$, respectively. (**D**) Cumulative probability histogram of spontaneous event rate per synapse per minute in 2 and 8 mM extracellular $Ca^{2+}$ solution containing folimycin (n = 175 synapses from four coverslips for 2 mM $Ca^{2+}$ and 216 synapses from four coverslips for 8 mM $Ca^{2+}$; $KS_{test}$ p < 0.05 Dmax = 0.8 fusion events per synapse per minute, with average rates of 0.4 ± 0.02 and 1.1 ± 0.02 (p value <0.05) fusion events per minute per synapse for 2 and 8 mM $Ca^{2+}$, respectively).

release sites did not show appreciable correlation (at 2 mM $Ca^{2+}$: slope = 1.0, $R^2$ = 0.08; at 4 mM $Ca^{2+}$: slope = −0.14, $R^2$ = 0.006; at 8 mM $Ca^{2+}$: slope = 0.47, $R^2$ = 0.02) (*Figure 6B–D*). Although, a large fraction of the nerve terminals (>70%) were capable of maintaining both evoked and spontaneous neurotransmission across all $Ca^{2+}$ concentrations, the rate of spontaneous transmission and the probability of successful AP-stimulated transmission are not correlated within a given release site. Furthermore, these data imply that the processes that control the kinetics of trafficking vesicles released at rest and those released in response to stimulation are distinct not only between synaptic terminals, but within a single synaptic terminal.

## Discussion

To visualize spontaneous exocytosis and endocytosis of single synaptic vesicles, we used a lentiviral system to express the pH-sensitive GFP (pHluorin) fused to the luminal domain of the vesicular glutamate transporter (vGlut1) in hippocampal neurons. In an earlier study, we employed the same approach to investigate trafficking of single synaptic vesicles that fuse in response to AP stimulation where the time of stimulation can be used to identify a successful fusion event (*Leitz and Kavalali, 2011*). For spontaneous vesicle fusion, however, such a time stamp does not exist; therefore, to confirm our findings using vGlut1-pHluorin, we also used dual color imaging with a red-shifted pHluorin variant (pHTomato) fused to the luminal domain of the synaptic vesicle protein synaptophysin (SypHTomato) and a green $Ca^{2+}$-sensitive probe (GCaMP5K) fused to the post-synaptic density protein 95 (PSD-95-GCaMP5K). In this setting, we could monitor presynaptic vesicle fusion and post-synaptic $Ca^{2+}$ entry upon NMDA receptor activation. We found that in both systems, synaptic vesicles that fuse spontaneously are retrieved and re-acidified much faster than their counterparts that fuse in response to stimulation. The rapid decay was not due to lateral diffusion of the probe because in the presence of folimycin, these events remained stable without decay and fluorescence accumulated in a stepwise fashion. It is also important to note that as these experiments were performed at room temperature, we expect that at higher, more physiological temperatures the rates of endocytosis and reacidification may be much faster than are reported here (*Renden and von Gersdorff, 2007*). Additionally, we emphasize that the data presented here does not imply that the postsynaptic responses arising from spontaneous and evoked vesicle fusion are necessarily different. However, earlier work from our group

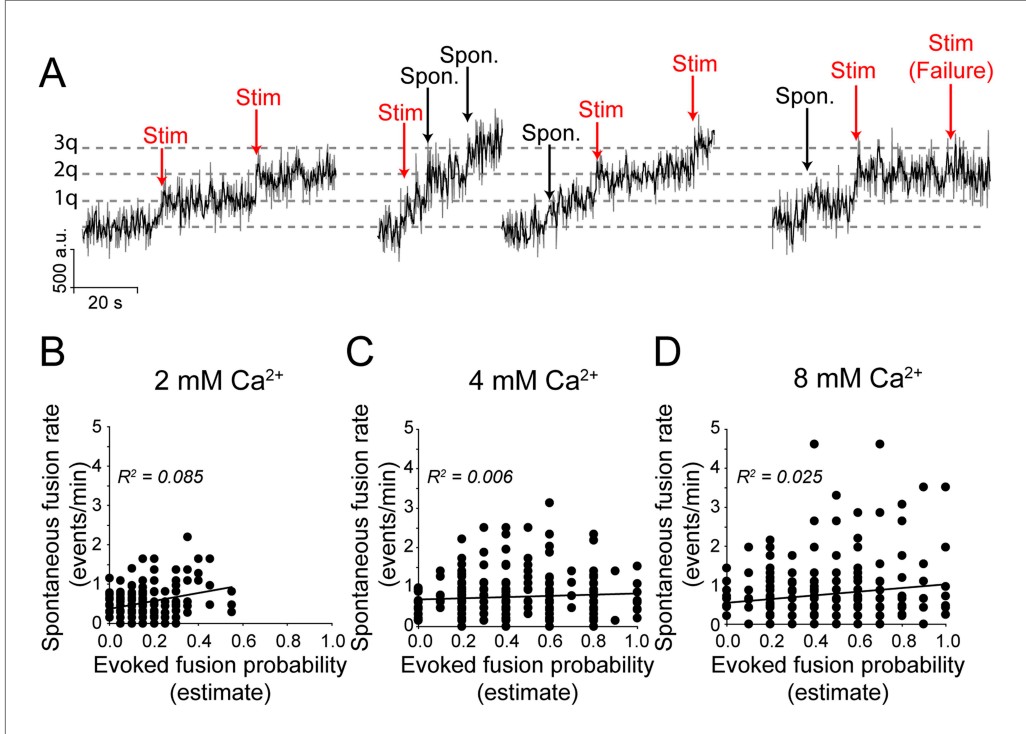

**Figure 6**. Spontaneous vesicle fusion rate and stimulation-evoked fusion probability do not correlate at a given synapse. (**A**) Example traces of fusion events in the presence of folimycin. Neurons in 2 mM $Ca^{2+}$ extracellular solution were stimulated with 1 AP delivered at 0.1 Hz while neurons in 4 and 8 mM $Ca^{2+}$ were stimulated with 1 AP delivered at 0.033 (30 s inter-stimulus interval) due to the higher probability of release. Fusion events were categorized as spontaneous or evoked by their temporal distance from the stimulation time, with events within ±1 s of stimulation selected as evoked fusion events. Note that spontaneous and evoked events are both quantal, indicated by the dashed grey line. (**B**) There is limited correlation between spontaneous fusion rate and evoked fusion probability in 2 mM $Ca^{2+}$. Distributions were best fit with a linear trend line with slope = 1.0 (Pearson's correlation coefficient $R^2$ = 0.08, p value <0.01 and Spearman r = 0.16, p value = 0.02; n = 200 synapses from four coverslips). (**C**) The correlation between spontaneous and evoked transmission decreases further as extracellular $Ca^{2+}$ increases to 4 mM best fit line with slope = 0.1 (Pearson's $R^2$ = 0.01, p value = 0.17 and Spearman r = 0.032, p value = 0.58; n = 311 synapses from six coverslips). (**D**) The correlation remains low between spontaneous and evoked transmission in 8 mM extracellular $Ca^{2+}$ with best fit line with slope = 0.5 (Pearson's $R^2$ = 0.02, p value = 0.045 and Spearman r = 0.066, p value = 0.35; n = 240 synapses from four coverslips).

(*Atasoy et al., 2008*; *Sara et al., 2011*) does suggest there is a physical segregation of postsynaptic receptor activation patterns in response to evoked and spontaneous fusion events. Unfortunately, here we cannot measure evoked quantal postsynaptic $Ca^{2+}$ signals during stimulation, as in our preparation, field stimulation itself causes direct postsynaptic effects. Therefore, in this study we did not examine the kinetic properties of evoked postsynaptic quantal events. Instead, we emphasize that the retrieval and reacidification of spontaneously fusing vesicles are distinct from vesicles that fuse in response to stimulation.

Unlike stimulation-evoked responses, these spontaneous increases in fluorescence decayed immediately without a detectable pause after the initial increase in fluorescence (dwell time), suggesting a minimal surface residency time of the pHluorin or pHTomato probes. Taken together, these results indicate that spontaneous synaptic vesicle exocytosis and endocytosis are tightly coupled processes and that the decay phase of these transients was mainly due to vesicle re-acidification upon endocytosis (*Alabi and Tsien, 2012*). Although we see extremely fast endocytosis of spontaneous recycling vesicles, large probes such as antibodies against the luminal domain of synaptotagmin1 or horseradish peroxidase are known to label spontaneously endocytosing vesicles (*Sara et al., 2005*; *Fredj and Burrone, 2009*). Therefore, spontaneously endocytosing vesicles may still form a fairly large fusion pore or may even fully collapse onto the plasma membrane—albeit without lateral dispersion of their

protein components—despite being retrieved quickly. However, we cannot fully exclude the possibility that some spontaneous fusion events may form narrow fusion pores where uptake of large probes can be curtailed as seen after spontaneous fusion of peptidergic vesicles (*Vardjan et al., 2007*).

The rate of endocytosis and vesicle reacidification found here is considerably faster than some of the previous reports that describe endocytosis with a time constant of ~14 s (*Granseth et al., 2006*; *Balaji and Ryan, 2007*) and vesicle reacidification rate of ~4 s (*Atluri and Ryan, 2006*). The discrepancy between our work and these earlier studies may be attributed several differences. First, our previous work has indicated that in response to increasing $Ca^{2+}$ concentrations as well as in boutons with higher release probability the dwell time and the decay time of single vesicle fusion events increase (*Leitz and Kavalali, 2011*). Therefore, we believe it is critical to select boutons with a wide range of release probabilities for single vesicle analysis. Our current work extrapolates this observation to spontaneous release events where we detect events with even faster kinetics. The vesicle re-acidification rates we estimate, on the other hand, are in agreement with studies using synapto-brevin-pHluorin (*Gandhi and Stevens, 2003*) as well as synaptophysin-pHluorin based measurements (*Zhang et al., 2009b*). Furthermore, it is important to note that some of these earlier studies were performed in extracellular solution containing 25 mM HEPES while here, and in our previous work (*Leitz and Kavalali, 2011*), we chose a lower HEPES concentration of 10 mM as HEPES is known to have phototoxic effects at higher concentrations (*Zigler et al., 1985*).

Why is synaptic vesicle re-acidification after spontaneous fusion faster than vesicle re-acidification after evoked fusion? We propose three non-mutually exclusive scenarios that can explain this difference. First, the pH buffering capacity of vesicles that endocytose after AP stimulation could be higher. Second, the function of the v-ATPase on vesicles that endocytose after AP stimulation may be slowed down since this complex is known to incorporate $Ca^{2+}$ sensor proteins (*Zhang et al., 2008*) and was recently shown to be differentially regulated during evoked vs spontaneous fusion (*Wang et al., 2014*). Finally, vesicles endocytosed during activity may rapidly trigger formation of larger vesicular structures that are expected to be slower to re-acidify due to their larger volume, consistent with findings from capacitance measurements in salamander photoreceptors (*Van Hook and Thoreson, 2012*) as well as recent electronmicrographic analysis of endocytosis in hippocampal synapses after rapid high-pressure freeze fixation (*Watanabe et al., 2014*).

In this study, we also investigated the $Ca^{2+}$-dependent regulation of spontaneous synaptic vesicle fusion events. At elevated $Ca^{2+}$ concentrations we detected an increase in the amplitudes of fusion events consistent with exocytosis of multiple synaptic vesicles. As noted above, at our resolution we cannot exclude the possibility that the observed increase in amplitude could be due to vesicle fusion from multiple release sites within a region of interest, especially if elevated $Ca^{2+}$ concentrations can activate adjacent release sites resulting in apparent multivesicular release albeit originating from two neighboring release sites. The discrepancy between the increases in mEPSC frequency we detect in electrophysiological recordings vs optical recordings of spontaneous events could indicate that some of the multivesicular events are occurring at neighboring synapses activated at elevated $Ca^{2+}$. However, our measurements of stimulation-evoked vesicle fusion probability at 2 mM Ca2+ are consistent with the dominance of single release sites in our analysis. Additionally, it is possible that increasing extra-cellular $Ca^{2+}$ specifically promotes the spontaneous fusion of synaptic vesicles containing elevated copy number of vGlut-pHluorin endowed by either a larger physical size of the vesicle or an elevated intrinsic copy number of vGlut-pHluorin. However, in part due to the similarity between our results and the bursting activity observed previously using single-synapse recordings in hippocampal neurons (*Abenavoli et al., 2002*); we consider the increase in fluorescence amplitude as arising from the rapid successive fusion of multiple vesicles as the most cell biologically parsimonious. If this interpretation is correct, we also found that fluorescence signals originating from these multivesicular events were not slower in their rate of decay, which contrasts our earlier observations on evoked multivesicular fusion events (*Leitz and Kavalali, 2011*), indicating that spontaneous exocytic load (i.e. the number of spontaneously fused vesicles on the plasma membrane) does not significantly impact retrieval kinetics. Alternatively, even if the larger amplitude spontaneous events were due to the fusion of enlarged synaptic vesicles or synaptic vesicles containing more fluorophore, this interpretation would still imply that the size of synaptic vesicles or the number of vGlut-pHluorin molecules within a synaptic vesicle (more membrane or more cargo, that is the exocytic load) do not impact their retrieval kinetics at rest.

In contrast to our earlier findings with vesicles that fuse in response to APs (*Leitz and Kavalali, 2011*), we found that the kinetics of endocytosis of spontaneous fusion events were not slowed in response

to elevated extracellular $Ca^{2+}$ concentrations. In all likelihood, at 8 mM $Ca^{2+}$ the average kinetics of fluorescence decay became faster due to emergence of events that could only be detected following inhibition of re-acidification. Therefore, with increasing extracellular $Ca^{2+}$ there was a clear increase in the rate of spontaneous fusion and the number of vesicles that undergo fusion. The relative insensitivity of vesicle retrieval kinetics to $Ca^{2+}$ suggests that synaptic vesicle retrieval after spontaneous and evoked fusion is regulated via diverse mechanisms consistent with differential dependence of the two forms of endocytosis on distinct dynamin isoforms (*Chung et al., 2010*; *Raimondi et al., 2011*; *Meng et al., 2013*) and distinct postendocytic transport machineries (*Peng et al., 2012*).

Finally, in our system we were able to directly compare the rate of spontaneous fusion and evoked-fusion probability within a single synapse. This analysis did not reveal a significant correlation between spontaneous fusion rate and evoked fusion probability estimates from individual release sites, further supporting the notion that these two modes of neurotransmission are controlled and maintained independently (*Sara et al., 2005*; *Atasoy et al., 2008*; *Fredj and Burrone, 2009*; *Melom et al., 2013*; *Peled et al., 2014*; *Wang et al., 2014* but see *Groemer and Klingauf, 2007*; *Wilhelm et al., 2010*). Importantly, at elevated $Ca^{2+}$ concentrations the propensity to fuse of each of the two forms of vesicle fusion was increased (*Figure 6*). However, consistent with their less steep dependence on $Ca^{2+}$, spontaneous fusion events showed a milder 1.5–2-fold increase compared to the threefold increase detected in the propensity of evoked fusion events. Nevertheless, their lack of correlation persisted suggesting a divergence in the mechanisms that regulate $Ca^{2+}$ sensitivity of evoked and spontaneous fusion events (*Xu et al., 2009*; *Groffen et al., 2010*; *Pang et al., 2011*; *Vyleta and Smith, 2011*; *Ermolyuk et al., 2013*). This difference in $Ca^{2+}$ regulation of spontaneous and evoked fusion probability may underlie differential sensitivity of the two forms of neurotransmission of certain neuromodulators and $Ca^{2+}$ signaling pathways (*Peters et al., 2010*; *Ramirez and Kavalali, 2011*; *Bal et al., 2013*). Recent studies in the Drosophila neuromuscular junction have shown that a substantial fraction of release sites carry out exclusively spontaneous or evoked neurotransmitter release (*Peled et al., 2014*; *Walter et al., 2014* also see ; *Melom et al., 2013*). In our measurements, we detected a substantial overlap of release sites that are capable of both forms of neurotransmission, in agreement with our earlier estimates from lower temporal resolution experiments (*Atasoy et al., 2008*). This discrepancy may be consistent with the premise that in immature presynaptic release sites, spontaneous neurotransmission dominates and release gradually shifts towards evoked transmission during synapse maturation (*Polo-Parada et al., 2001*; *Mozhayeva et al., 2002*; *Andreae et al., 2012* also *Walter et al., 2014*). Taken together, the findings we present here provide insight into the segregation of spontaneous and evoked neurotransmitter release mechanisms at the level of single synaptic vesicle fusion events. The differential regulation of spontaneous vesicle fusion suggests it has a role in neuronal signaling distinct from information transfer patterns mediated by evoked release, even within a single synapse.

## Materials and methods

### Cell culture

Dissociated hippocampal neurons were cultured from postnatal day 0–3 Sprague Dawley rats of either sex as described previously (*Kavalali et al., 1999*). At 4 days in vitro (DIV), cultures were infected with lentivirus expressing vGlut-pHluorin or with SypHtomato and PSD-95-GCaMP, and experiments were conducted between 15–20 DIV when synapses reach maturity (*Mozhayeva et al., 2002*). All experiments were performed at room temperature.

### Lentiviral preparations

In these experiments we relied on a lentiviral expression system. The vGlut-pHluorin construct was a generous gift from Drs Robert Edwards and Susan Voglmaier (University of California, San Francisco). A modified, synaptophysin pHTomato was a generous gift from Dr Richard Tsien (New York University Medical Center and Stanford University). The primers:

ATATggatccggtggttctggtgtgagcaagggcgaggagaataacatggccatcatcaaggagttcatgcgcttcaag (pHTomato.FIX.F) and atataccggtaccagaaccacccttgtacagctcgtccatgccgccggtggagtggcggccc (pHTomato.FIX.R) were used to return pHTomato to the published version and attach small flexible linkers. All lentiviruses were prepared by transfection of human embryonic kidney (HEK) 293-T cells with the plasmid of intrest together with viral coat and packaging protein constructs (pVSVG, pRsv-Rev, and pPRE) using FuGENE 6 (Promega, Madison, WI). 3 days after transfection, virus was harvested from HEK 293-T cell-conditioned media and added to neuronal media at 4 DIV.

## vGlut1-pHluorin imaging and analysis

Single-wavelength experiments were performed using an Andor iXon Ultra 897 back-illuminated EMCCD camera (Model no. DU-897U-CSO-#BV) collected on a Nikon Eclipse TE2000-U microscope with a 100X Plan Fluor objective (Nikon). For illumination we used a Lambda-DG4 (Sutter instruments, Novato, CA) with FITC filter. Images were acquired at ~8 Hz with an exposure time of 100 ms and binning 4 (Using Nikon Elements Ar software). For analysis, square regions of interest (ROIs) with length and width of 2.5 μm were generated and the resulting fluorescence values were exported to Microsoft Excel for analysis. Successful fusion events were those where the average of 3 points was greater than twice the standard deviation of 17 points (~2.1 s) prior provided the recordings were stable for at least 78 frames (~10 s) preceding the increase in fluorescence. Additionally, at least one additional point after the initial fluorescence increase had to be greater than twice the standard deviation of the 17 points prior; this excluded large single point increases in fluorescence from which decay times could not be determined. Noise measurements were obtained in an analogous fashion: by subtracting the average of 3 points from the mean of 17 points prior during a random period during sampling. Decay times were analyzed in Clampfit (Molecular Devices, Sunnyvale, CA) by fitting raw data with a single exponential decay using Levenberg–Marquardt least sum of squares minimizations. For all control experiments, the extracellular solution was a modified Tyrode's solution containing (in mM): 150 NaCl, 4 KCl, 10 glucose, 10 HEPES, 2 MgCl and varying concentrations of $CaCl_2$ (2, 4, and 8 mM), pH 7.4 (310 Osm). In experiments with folimycin (Concanamycin A, Sigma, St. Louis, MO), a final concentration of 80 nM was used. For Tris-buffered experiments, solutions contained (in mM): 108 NaCl, 4 KCl, 2 $MgCl_2$, 10 glucose and 50 Tris–HCl and 2 or 8 $CaCl_2$. For all Tris buffering experiments, neurons were allowed to equilibrate in extracellular solution for at least 10 min prior to imaging. We elected to use a Tris-buffered extracellular solution because it does not require continuous bubbling during perfusion; a prerequisite of bicarbonate based buffers. Moreover, it has been shown that incubation in Tris can buffer, and thus slow, vesicle reacidification (*Gandhi and Stevens, 2003*; *Zhang et al., 2009a*). All solutions were adjusted to pH 7.4 with NaOH and 310 Osm prior to use. To prevent network activity, postsynaptic ionotropic glutamate receptor antagonists 6-cyano-7-nitroquinoxaline-2,3-dione (CNQX 10 μM; Sigma) and D(−)-2-Amino-5-phosphonopentanoic acid (AP-5 50 μM; Sigma) were added to the experimental solutions. To prevent spontaneous action potential generation, we incubated neurons in 1 μM tetrodotoxin (TTX). For simultaneous spontaneous activity and evoked stimulation experiments, TTX was omitted from the extracellular solution and neurons were stimulated using parallel bipolar electrodes (FHC) delivering 15–20 mA pulses with pulse width of 1 ms. At the end of each experiment, Tyrode containing 20 mM $NH_4Cl$ was added to help identify putative synaptic boutons. Due to the lower intrinsic release probability in 2 mM extracellular $Ca^{2+}$, we gave twice as many stimulations (20 APs delivered 15 s apart) as in 4 and 8 mM extracellular $Ca^{2+}$ (10 APs delivered 30 s apart), thus evoked fusion probability estimates in 2 mM $Ca^{2+}$ are in multiples of 0.05 and not 0.1.

## SypHTomato/PSD-95-GCaMP imaging and analysis

For dual color experiments, FITC and TRITC filters (Chroma Technology, Bellows Falls, VT) were inserted into the Lamda-DG. Images were collected on the same hardware as above but with increased acquisition interval of 180 ms (using 40 ms FITC and 140 ms TRITC excitation intervals). The extracellular solution was the same as above, however 2,3-dihydroxy-6-nitro-7-sulfamoyl-benzo[f]quinoxaline-2,3-dione (NBQX 10 μM; Tocris, UK) was used instead of CNQX as it is a more specific AMPAR antagonist. AP-5 and $Mg^{2+}$ were omitted from all experiments to allow $Ca^{2+}$ entry through NMDARs. Full waveforms of sypHTomato and GCaMP5K fluorescence were fit with a double exponential decay time and linearized to correct for photobleaching. Corrected fluorescence waveforms were analyzed as above for fluorescence increases. Once fusion events were identified, the raw data at that time was linearly corrected according to the baseline slope from 15 s before the event. This method proved successful as in the absence of folimycin events would decay back to baseline but in the presence of folimcyin the events had a staircase fluorescence waveform. All sypHTomato and GCaMP5K events were then aligned and averaged for each experiment.

## Multivesicular delay analysis

Hypothetical single-vesicle fusion events were generated in Microsoft Excel with amplitudes of 623 ± 122 a.u and decaying as a single exponential with decay time of 371 ms (as determined by the non-geometric average decay time of events in 2 mM $Ca^{2+}$ fit using a Beta-function). 4,000 hypothetical

fusion events were then generated with varying amplitudes using the random number generator in Microsoft Excel with constraints of a Gaussian distribution according the first Gaussian curve (1q = 623 ± 122 a.u) in 2 mM $Ca^{2+}$. Random pairs of these amplitudes were then summed together with varying temporal offsets (delays between the first and second hypothetical events) to generate 2000 hypothetical 2-vesicle fusion events. The amplitudes of the three point moving averages were then calculated—analogous to the amplitude calculations of raw data—and the distribution of these hypothetical multive-sicular events was compared to the multivesicular only component of the amplitude distribution in 8 mM extracellular $Ca^{2+}$. The most similar distribution had a temporal offset of 118 ms, which we interpret as an approximate interval between two vesicles fusing spontaneously during multivesicular fusion.

## Statistics

Statistics were performed using Microsoft Excel or GraphPad Prism 6. For statistical comparisons between experiments that were performed on the same population of boutons, Student's t-test was used (2-tailed, paired). In imaging experiments, n refers to the number of multiple experiments performed, with each experiment containing up to 100 regions of interest. Student's t-test (2-tailed, unpaired) was used to analyze all pair wise data sets obtained from boutons under distinct conditions. The Kolmogorov–Smirnov test was used to determine differences in cumulative probability distributions. The chi-square test was used to evaluate the goodness fit of Gaussian distributions to amplitude histograms. For analysis of multiple comparisons among synaptophysin-pHtomato experiments one-way ANOVA with Bonferroni post hoc analysis was used. For the correlation analysis, both parametric and non-parametric correlation coefficients were determined using GraphPad Prism 6. Linear regression analysis included a test for the hypothesis that the slope = 0, again using GraphPad Prism 6. Finally, the D'Agostino-Pearson omnibus K2 normality test was performed in GraphPad Prism 6.

## Acknowledgements

We thank members of the Kavalali and Monteggia laboratories, in particular Drs. Lisa Monteggia, Elena Nosyreva, Denise Ramirez and Devon Crawford, as well as Drs. Robin Hiesinger, Helmut Kramer and Donald Hilgemann for insightful discussions and comments on the manuscript. This work was supported by NIH grants MH066198 (E.T.K) and the Cellular Biophysics of the Neuron Training Program T32 NS069562 (J.L.).

## Additional information

### Funding

| Funder | Grant reference number | Author |
| --- | --- | --- |
| National Institute of Mental Health | R01 MH066198 | Ege T Kavalali |
| National Institute of Neurological Disorders and Stroke (NINDS) | T32 NS069562 | Ege T Kavalali |

The funders had no role in study design, data collection and interpretation, or the decision to submit the work for publication.

### Author contributions

JL, Conception and design, Acquisition of data, Analysis and interpretation of data, Drafting or revising the article; ETK, Conception and design, Analysis and interpretation of data, Drafting or revising the article

### Ethics

Animal experimentation: All animal protocols were approved by the Institutional Care and Use Committee at UT Southwestern Medical Center. The work presented in this study is covered by the Animal Protocol Numbers APN 0866-06-05-1 and APN 0866-06-03-1

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
