## [Decision Letter]

Thank you for sending your work entitled “Fast retrieval and autonomous regulation of single spontaneously recycling synaptic vesicles” for consideration at *eLife*. Your article has been favorably evaluated by Eve Marder (Senior editor) and 3 reviewers, one of whom is a member of our Board of Reviewing Editors.

The Reviewing editor and the other reviewers discussed their comments before we reached this decision, and all felt that this work is interesting, providing novel approaches and insights in vesicle exocytosis.

The reviewers felt that the following major points of analysis and interpretation are however needed to be adequately addressed to publish at *eLife*.

1) Quantification. How tightly linked are simultaneously occurring pre- (single vesicle release) and post-synaptic (Ca imaging) spontaneous events? Only postsynaptic events that follow immediately after presynaptic events are of relevance.

2) How well can different release sites (or nearby boutons) be separated? Are multiple vesicles being released from one bouton within 120 ms (if so, please do not call this multiquantal release, which is more in the time scale of one's and tens of millisec: that is confusing)?

3) Are events from spontaneous and evoked release really different? One potential test: Do the evoked release events also correlate with a Ca signal that is driven by NMDARs?

4) There were also significant concerns about the histogram analysis and multiquantal release conclusion. It was unclear why a fixed delay between two sequential release events was used while alternative hypothesis (that these events are just a bit larger and the distribution is not exactly normal) was not considered simpler? At least the alternative interpretation should be discussed. Also, it is essential that the fusion frequencies (Figure 5) are recalculated using the actual number of detected events (regardless of amplitude), so that the frequencies are not dependent on a specific interpretation such as multivesicular release.

Please find below the initial reviewer comments that led to the consensus view above. There are additional issues in these reviews that should also be addressed.

Reviewer #1:

This is an excellent and very interesting paper that reveals several striking differences between spontaneous and evoked vesicle retrieval and reacidification rates. The data analysis is convincing and carefully performed. A surprisingly fast rate of vesicle endocytosis and reacidification is shown after spontaneous fusion events. The results are of fundamental importance for an understanding of CNS synaptic transmission. Together, these results present a strong case for major differences between spontaneous and evoked vesicle recycling and acidification rates.

Major:

1) Results section: For the average decay time constant (e.g. 3.7s) and median decay time (e.g. 0.28) please add an error bar to these numbers as you have done elsewhere in the Results section.

2) Results section: For how long are the neurons bathed in the 50 mM TRIS solution before experiments are performed?

3) In Figure 5 what was the average frequency of the spontaneous fusion events in 2 mM Ca and in 8 mM Ca? Please give the average number with error bar. It seems like a 3-fold increase in frequency. Also, what is the average increase in spontaneous mEPSC frequency from 2 mM external Ca to 8 mM external Ca? This seems like an easy experiment for the authors to conduct and it would be very interesting to compare the numbers of this change to the numbers from the imaging data.

4) Change the first paragraph of the Discussion into three paragraphs.

5) Why not try the experiment in Figure 1 with an external solution that is based on 25 mM bicarbonate (a more physiological pH buffer)? Is there a problem with exchanging the HEPES-based solution of the cultures with a bicarbonate-based solution and then performing the experiment of Figure 1? A vesicle that recycles with 10 mM HEPES in its lumen may have a different pH buffering than one with 25 mM bicarbonate. If an experiment cannot be performed with bicarbonate then some discussion of this issue should be given in the text.

6) What is the CV of the distributions in Figure 1? Seems like a CV of about 0.4-0.5, which is close to that of mEPSCs. It would be interesting to know how well these match each other.

Reviewer #2:

The authors use Phluorin imaging to detect spontaneously fusing vesicles and determine their reacidification kinetics, as well as their calcium-dependence. The study is well performed, the detection of spontaneous release using pHluorin constructs is impressive and overall believable, and the authors reach conclusions, which are important and interesting for the field. However, there are some points that require clarification, which will include more analysis, and the interpretation regarding multivesicular release appears not sufficiently supported (details below).

Major points:

1) A major point is that it is not clear that the optically detected 'spontaneous events' really represent fusion of synaptic vesicles, and not for instance fusion of endosomes. The authors in principle address this point elegantly by co-expressing a fluorescence calcium-indicator (GCaMP5K) fused to PSD-95 and doing dual-color imaging. This is a nice approach, but from the text and analysis presented it does not become clear whether there was a tight correlation between pre- and post-synaptic events, and therefore there is not enough evidence for the conclusion: “...spontaneous fusing vesicles elicit postsynaptic Ca2+ signals”. For instance in Figure 2: are those events (red and green channel) detected simultaneously? How often did spontaneous increases in the red channel correlate with increases in the green channel? How many events in the red channel did not coincide with events in the green channel? The authors could calculate and plot the waiting times between sequential red and green events at identified synapses. If the authors are right, such a plot should have a peak at very short intervals. Finally, the authors write “Spontaneous increases in fluorescence that were correlated (within {plus minus} 1 s) with Ca2+ signals..” I do not understand the {plus minus} here: presumably the relevant events would be those were the red event would precede (or coincide with) the green event, not the other way around. Overall, the authors need to present additional analysis of these data.

2) I am skeptical about the conclusion that the larger amplitudes of events at higher calcium concentrations are due to multivesicular release (Figure 3, text in the Results section). The larger events in the presence of higher calcium concentration could be because of the preferential fusion of slightly larger vesicles under these circumstances (or even due to a change in the photophysical properties of pHluorin). The author's argument doesn't make sense to me: “It is unlikely that this increase in fluorescence is due to variation in the number of pHluorin molecules, as the increase in amplitude is observed when extracellular Ca2+ concentrations are increased.” Why not? The histogram of event sizes could not be fitted with a single Gaussian, but who is to say whether spontaneously fusing vesicles exactly follow a Gaussian distribution? The three distributions at 1q, 1.3q and 2q are not visible as peaks in the histogram at all; especially the fit of a distribution at 2q appears unreasonable. Finally, the suggestion that events with normalized amplitude at 1.3 q would result from multivesicular release of two vesicles with a fixed delay (118 ms) appears unreasonable; what mechanism would ensure that multivesicular release would happen always with this delay, especially as this is spontaneous release? I think the authors should remove this interpretation, which has consequences also for the interpretation of Figure 4 and for the rates reported in Figure 5 and possibly in Figure 6.

3) The authors use the lack of a correlation between decay time and amplitude as an argument that “there is a fundamental difference in the kinetics of endocytosis and reacidification between vesicles that fuse spontaneously and those that fuse in response to stimulation.” But I would argue that this finding, together with the previous work of the authors, [25] is an argument against the interpretation of multivesicular release.

4) To follow up on the last statement above in point 2, to estimate the fusion frequencies (reported in Figure 5) the authors “counted all events with amplitudes within the first quantal mean as a single event, and events with larger amplitudes as two events”. This is not appropriate, first because it is not clear whether larger events represent multiple events, and second because the distributions at 1q and 1.3q overlap. The authors should only use the number of events that they can detect as such, independently of amplitude. The same goes for Figure 6, if the rates were calculated in the same way.

5) The correlation analysis in Figure 6 has not been described in the statistics section of the Materials and Methods. Was this Pearson's correlation coefficient? The R^2 is quite small, but nevertheless the slope is positive under all circumstances. The authors should add a statistical test of the hypothesis R=0, and report whether there is a significant (but small) correlation. Finally, the authors should explain why all evoked fusion probabilities are integer multiples of 0.1 (panel C, D), or 0.05 (panel B). Why this difference between the conditions? Finally, given that fusion probabilities can be measured in only 10 (C, D), or 20 (B) discrete categories, would non-parametric correlation analysis be more appropriate?

Reviewer #3:

In the current manuscript, “Fast retrieval and autonomous regulation of single spontaneously recycling synaptic vesicles,” Leitz and Kavalali investigate the retrieval and re-acidification of vesicles spontaneously released, and find that these events have a faster decay than single vesicles released with AP stimulation. Using dual color imaging, they also show that spontaneous events can elicit NMDAR-mediated calcium responses. Additionally, they show that multiple vesicles can be release spontaneously at a single release site, and that this occurs more frequently with increasing calcium concentrations, without showing a change in the pHluorin decay kinetics. They argue that the different release modes, spontaneous and evoked, may use distinct methods of endocytosis. Furthermore, as they show there is no correlation between the release probability of evoked events and the frequency of spontaneously released events at single release sites, they conclude that these are two functionally distinct vesicle pools.

Overall, this is a well-written and clear paper with many appropriate controls. The dual color imaging of the spontaneous release events and postsynaptic response are important and interesting, while some concerns exist and are discussed below.

1) Detection of single, spontaneous release events is a crucial technique for this paper. Though the amplitude distribution of the detected events looks convincing, it would also be useful to show how the distribution shifts when spontaneous release is blocked (i.e. with tetanus toxin incubation). Additionally, it is not clear how the noise events were detected. It would also be necessary to show how what the detection criteria discover when run with a negative amplitude (i.e. -2X the SD of the 17 points prior to the event).

2) For Figure 2, the correlation between the spontaneous pHluorin events and spontaneous Ca2+ events was stated to be {plus minus} 1 s. While the limitations for time resolution in the dual color imaging are expected, since the vesicle release should directly cause the Ca2+ influx, these events should really only be considered if the Ca2+ influx occurs in a time range after the detected pHluorin event.

3) Figure 2. What do the corresponding Ca2+ signals look like for the evoked SypHTomato events? Can they be more closely time-locked?

4) Figure 3. Multivesicular release events recorded by electrophysiology are thought to occur essentially simultaneously. The 1.3q events seem likely to be events happening with a 118 ms delay. Also to this point, how well resolved are the single release sites? Could the multiple vesicle release events be closely neighboring synapses, which happen to release vesicles in close temporal proximity? Also, are the multivesicular spontaneous events Ca2+ dependent? How do they look in the presence of Cd2+?

5) Figure 5. The frequency of spontaneous release events clearly increases in 8 mM Ca2+ in the presence of folimycin. The authors explain this as recruitment of very rapidly retrieved vesicles. However, does the frequency of spontaneous events also increase in folimycin with 2 mM Ca2+? From the Figure, it doesn't appear to, but it would be interesting to see.

6) Figure 6. This is an important figure for the authors' interpretation of the separate pools of vesicles (spontaneous and evoked) released with distinct properties at single synapses. However, a few issues could cloud the interpretation from the lack of correlation between evoked fusion probability and spontaneous release frequency. First, for an event to be considered evoked with a 1 s lag time from the stimulus seems too long even for asynchronous release with a single action potential. Second, there is again the issue of spatial resolution. Are these actually single release sites? Finally, the release probability (the calculation of which should be clearly stated in the text) seems very high for these synapses in 4 mM Ca2+. This could argue for the multiple release site measurement.

7) Figure S1. The increase in signal for the PSD-95-GCaMP5K is difficult to interpret, because it could be due to either the increased extracellular Ca2+ or due to the increased multivesicular release events. This could also be interesting to see in the presence of Cd2+.

[Editors’ note: further revisions were requested before acceptance, as shown below.]

Thank you for resubmitting your work entitled “Fast retrieval and autonomous regulation of single spontaneously recycling synaptic vesicles” for further consideration at *eLife*. Your revised article has been favorably evaluated by Eve Marder (Senior editor) and the original three reviewers. The manuscript has been improved but there are some remaining issues that need to be addressed before acceptance, as outlined below:

All three reviewers would like to see that all (new) analysis is properly implemented in the manuscript. Specifically, there are arguments that you made in your rebuttal letter that do not appear in the manuscript, but should. Moreover the reviewers generally all agree that support for multivesicular release is not that strong, and feel that this claim should be better balanced by dealing with caveats and alternative explanations.

Additional comments that surfaced in response to your revision are provided below:

1) Regarding detection of events, the authors added improved analysis. One reviewer would like to see addition of “false positive” events either from negative implement the detection (inversed) or from 0 Ca2+ to the histograms for amplitude and decay times. In addition, it would be helpful to subtract weighted average decay of false positives from weighted average decay of spontaneous events to obtain “net events”.

2) Coincidence of pre- and postsynaptic events: The detection of postsynaptic responses at the -1 frame will still be confusing to the reader, and additional clarification in the text is required. Given that the distribution of events puts most of them between 0 to 5 frames after stimulation, this is the time window that the authors should use for analysis. They will lose very few events and it will be more straightforward as it makes little sense that the postsynaptic signal should occur prior to the presynaptic signal. If the -1 frame events are to be included, the explanation for why these events would appear to occur before the stimulation, as included in the rebuttal letter must also be in the text.

Unfortunately, the postsynaptic signals coupled to the evoked Syp-Tomato signals are missing, and this is still an obvious gap in the paper. As the authors explain in the rebuttal letter, this is due to an understandable technical issues with field stimulation and network activity, although that the field stimulation itself causes a signal is strange when NMDARs are not blocked by Mg2+. Again, the authors should address this issue in the text. Pointing out to the reader that the recording the postsynaptic events with stimulation is technically difficult does not take anything away from the findings, and will also preempt questions from a reader who sees the comparison of the postsynaptic signals generated by evoked or spontaneous release as the obvious next step.

3) Release probability vs spontaneous release: There is only a 50% increase in the average spontaneous fusion rate with Ca2+ increase to 8 mM, while average release probability increases 3-fold. This figure could simply suggest that the release frequency of minis is much, much less sensitive to calcium than evoked release probability. This should be discussed by the authors.

4) Sensitivity to detect individual events/multiquantal release: Overall, the single site, multiquantal spontaneous release claim is still not convincing (or at least multiple release site cannot be ruled out convincingly). I do not understand how the [31] citation excludes the possibility that the authors are not detecting coincident release from neighboring synapses and not the multi-quantal events they report-particularly as the release probability they claim will result in individual sites in 2mM Ca2+ and this will be less likely occur as Ca2+ increases. Additionally, aside from the increase in release probability with increased calcium, other factors that contribute to ability to differentiate individual synapses include the density of synapse and the amount of binning.

The discrepancy between mEPSC frequency increase in electrophysiological recordings and optical recordings of spontaneous events (though small) could suggest the events reported as multiquantal are, in fact, neighboring synapses. It would be worth testing when the portion of the population of events with 1.3 q and 2 q amplitudes are treated as individual events from neighboring synapses, does the frequency increase of the optical recordings better match the electrophysiological ones.

---

## [Author Response]

We would like to thank the reviewers for their insightful and constructive evaluation of our manuscript. In the revised paper, we have included new data and analysis. We now report the false positive detection rate and show the amplitude distributions of false positive events detected during minimal spontaneous vesicle fusion in nominal calcium concentrations (Modified Figure 1). Additionally, we now report the temporal correlation of postsynaptic Ca2+ entry reported by PSD-95-GCaMP5K and the presynaptic vesicle fusion reported by SypHTomato. We now include multiple interpretations for the observed Ca2+ increase in fluorescence amplitude and we now show the frequency of spontaneous fusion with and without interpreting increased fluorescence amplitude as multivesicular fusion (New Figure 5). The new experiments and analysis are briefly outlined below.

A) To estimate our false positive event detection rate, we re-analyzed our data using the inverse of the same detection criteria (-2X the standard deviation of the mean) we used previously to identify positive spontaneous vesicle fusion events. In this setting, we detected an average of 0.065 “negative” events per bouton per minute (compared to “positive” 0.5 events/bouton/min., see Figure 5). This measurement indicates that false positive hits impart a negligible contribution to our data set as a whole.

Additionally, we performed experiments in extracellular solution lacking Ca2+ to minimize the rate of vesicle fusion but including folimycin (to increase our confidence in putative positive events) to further validate our criteria to identify spontaneous fusion events. Here, we used our earlier detection criteria to identify increases in fluorescence that may reflect vesicle fusion. In nominal Ca2+, positive bona fide fusion events (detected in folimycin) were decreased to 0.01 events/bouton/min (compared to 0.5 events/bouton/min in 2 mM Ca2+).

We found that in this setting, transient increases in fluorescence (indicating possible false positives) had a reduced amplitude compared to bona fide spontaneous fusion events in 2 mM extracellular Ca2+ (New Figure 1). Moreover, when our detection criteria were set to identify negative increases in fluorescence amplitude, the absolute value of these -2X events were not different from the false positive events. Together these data validate our earlier detection criteria and indicate that our false positive detection rate is ∼ 10% per recording.

B) We compared the frame number difference between spontaneous increases in SypHTomato and PSD-95-GCaMP5K increases we found that 69% of spontaneous increases in SypHTomato elicited a PSD-95-GCaMP5K signal within ± 1 frame (±200 ms). This percentage increased to 83% between -1 frame to +5 frames. We have added these findings in the text.

C) We have analyzed the spontaneous vGlut-pHluorin fusion frequency data in two ways; independent of fluorescence amplitude, and dependent of fluorescence amplitude (new Figure 5). We have now included alternate hypotheses (preferential fusion of larger vesicles or non-normal distribution of event amplitudes possibly due to uneven distribution of fluorophores) in the text to better address the observed Ca^2+^-dependent increase in fluorescence amplitude.

D) Finally we have added additional non-parametric statistical analyses (e.g. Spearman’s Correlation) to the spontaneous fusion rate and evoked fusion probability correlation plots. This further analysis showed that there is only extremely modest correlation between the two rates at 2 and 8 mM Ca2+ concentrations whereas no detectable correlation at 4 mM Ca2+.

*1) Quantification. How tightly linked are simultaneously occurring pre- (single vesicle release) and post-synaptic (Ca imaging) spontaneous events? Only postsynaptic events that follow immediately after presynaptic events are of relevance*.

Below, we include a temporal correlation histogram of GCaMP5K event time (in frames) relative to SypHTo fusion (0 on the x axis). 67% of GCaMP events are within plus or minus 1 frame of the SypHTo event, increasing to 84% of events that occur from -1 to + 5 frames (∼1 sec). We include this -1 group because in our system we obtain only one time point for each cycle of imaging (thus one frame of both red channel and green channel constitute one time point). Therefore, fusion events that may occur halfway through an imaging cycle would appear shifted by -1 frame. Reanalysis of our data taking into consideration only these events that occur between -1 and +5 frames did not alter our previously reported data. We now also refer to these data in the text.Author response image 1.

2) How well can different release sites (or nearby boutons) be separated? Are multiple vesicles being released from one bouton within 120 ms (if so, please do not call this multiquantal release, which is more in the time scale of one's and tens of millisec: that is confusing)?

We cannot discriminate between multiquantal fusion and near-simultaneous fusion from nearby release sites. However, our evoked fusion probability estimates are consistent with earlier measurements of vesicle fusion probability using FM1-43 uptake into individual release sites under similar conditions (e.g. [31]). We now added this point to the text in the Results and in the Discussion.

3) Are events from spontaneous and evoked release really different? One potential test: Do the evoked release events also correlate with a Ca signal that is driven by NMDARs?

We do not mean to imply that the postsynaptic responses arising from spontaneous and evoked vesicle fusion are necessarily different. Earlier work from our group (6; 45) suggested a physical segregation between postsynaptic receptor activation patterns in response to evoked and spontaneous fusion events. However, so far, we did not detect any difference between the kinetic properties of evoked and spontaneous postsynaptic quantal events. Instead, in this study, we emphasize that retrieval and reacidification of spontaneously fusing vesicles are distinct from those of stimulation-evoked fusion vesicles. Unfortunately, we cannot measure quantal postsynaptic Ca2+ signals during stimulation, as in our preparation field stimulation itself causes direct postsynaptic effects. However, we are working to improve these recording conditions in our studies for the near future.

*4) There were also significant concerns about the histogram analysis and multiquantal release conclusion. It was unclear why a fixed delay between two sequential release events was used while alternative hypothesis (that these events are just a bit larger and the distribution is not exactly normal) was not considered simpler? At least the alternative interpretation should be discussed. Also, it is essential that the fusion frequencies (*Figure 5*) are recalculated using the actual number of detected events (regardless of amplitude), so that the frequencies are not dependent on a specific interpretation such as multivesicular release*.

We agree with the reviewers that our multiquantal release conclusion is one of many interpretations to our data. In our estimations we used a fixed delay to provide a proof of principle argument that multiquantal release could account for these observations. Although, we could test additional delays and kinetic parameters, given our limited resolution of these events, we prefer to refrain from over interpreting of our data. We now include a discussion of these alternative interpretations of the data in the text. We now report spontaneous fusion frequency estimates with and without taking into account the potential multiquantal nature of the events (see revised Figure 5).

Reviewer #1:

*1) Results section: For the average decay time constant (e.g. 3.7s) and median decay time (e.g. 0.28) please add an error bar to these numbers as you have done elsewhere in the Results section*.

We have now added error bars for these averages. Originally, we did not include error bars to these numbers because individual event parameters were not normally distributed. The non-geometric mean decay time constants were derived from fitting with β-distributions and instead we reported the upper and lower bounds of the distribution in the figure legends. Therefore, we have now included both geometric and non-geometric means in the text.

2) Results section: For how long are the neurons bathed in the 50 mM TRIS solution before experiments are performed?

We have added the duration of pre-incubation (10 min) to the Methods section.

*3) In*
Figure 5
*what was the average frequency of the spontaneous fusion events in 2 mM Ca and in 8 mM Ca? Please give the average number with error bar. It seems like a 3-fold increase in frequency. Also, what is the average increase in spontaneous mEPSC frequency from 2 mM external Ca to 8 mM external Ca? This seems like an easy experiment for the authors to conduct and it would be very interesting to compare the numbers of this change to the numbers from the imaging data.*

We have added the average frequency of spontaneous fusion events into the text. When we compare mEPSC rates at 2 mM Ca^2+^ and 8 mM Ca^2+^ we detect a ∼3.7-fold increase. In our optical experiments, we detect a ∼2.7-fold increase. Although the increase in spontaneous fusion rate detected by these two methods are in broad agreement (as we now indicated in the text), we would like to refrain from a detailed comparison at this time in the current manuscript to avoid over-interpreting our results. A reliable comparison of optical/electrophysiological data is constrained by the fact that the two measures are not obtained simultaneously and electrophysiological readout of distal synapses may be altered electrotonically.

4) Change the first paragraph of the Discussion into three paragraphs.

We agree and we have made this change.

*5) Why not try the experiment in*
Figure 1
*with an external solution that is based on 25 mM bicarbonate (a more physiological pH buffer)? Is there a problem with exchanging the HEPES-based solution of the cultures with a bicarbonate-based solution and then performing the experiment of*
Figure 1*? A vesicle that recycles with 10 mM HEPES in its lumen may have a different pH buffering than one with 25 mM bicarbonate. If an experiment cannot be performed with bicarbonate then some discussion of this issue should be given in the text.*

For practical reasons we chose to use Tris as a buffer. Bicarbonate buffers require constant carbogen bubbling during perfusion, which we currently could not implement on our imaging set up. We opted to use Tris as it has been previously used effectively to buffer synaptic vesicle pH ([16], Zhang et al., 2009).

*6) What is the CV of the distributions in*
Figure 1*? Seems like a CV of about 0.4-0.5, which is close to that of mEPSCs. It would be interesting to know how well these match each other*.

We have added the CV of our amplitude distributions (Figure 1 legend). However we caution comparing the two as the CV for the optical experiments is dependent on the number of pHluorins per vesicle (in addition to the number of vesicles) while the CV for mEPSCs depends on the number of postsynaptic receptors present and electrotonic filtering (besides the number of vesicles), and thus may be susceptible to different degrees of variance.

Reviewer #2:

*1) A major point is that it is not clear that the optically detected 'spontaneous events' really represent fusion of synaptic vesicles, and not for instance fusion of endosomes. The authors in principle address this point elegantly by co-expressing a fluorescence calcium-indicator (GCaMP5K) fused to PSD-95 and doing dual-color imaging. This is a nice approach, but from the text and analysis presented it does not become clear whether there was a tight correlation between pre- and post-synaptic events, and therefore there is not enough evidence for the conclusion: “...spontaneous fusing vesicles elicit postsynaptic Ca2+ signals”. For instance in*
Figure 2*: are those events (red and green channel) detected simultaneously? How often did spontaneous increases in the red channel correlate with increases in the green channel? How many events in the red channel did not coincide with events in the green channel? The authors could calculate and plot the waiting times between sequential red and green events at identified synapses. If the authors are right, such a plot should have a peak at very short intervals. Finally, the authors write “Spontaneous increases in fluorescence that were correlated (within {plus minus} 1 s) with Ca2+ signals..” I do not understand the {plus minus} here: presumably the relevant events would be those were the red event would precede (or coincide with) the green event, not the other way around. Overall, the authors need to present additional analysis of these data*.

We have corrected the text to state that we are looking at events that occur within -1 to +5 frames of the SypHTo signals. We found that the majority (84%) of events occurred within this time. Please refer to the answer we provided above for further details.

*2) I am skeptical about the conclusion that the larger amplitudes of events at higher calcium concentrations are due to multivesicular release (*Figure 3*, text in the Results section). The larger events in the presence of higher calcium concentration could be because of the preferential fusion of slightly larger vesicles under these circumstances (or even due to a change in the photophysical properties of pHluorin). The author's argument doesn't make sense to me: ”It is unlikely that this increase in fluorescence is due to variation in the number of pHluorin molecules, as the increase in amplitude is observed when extracellular Ca2+ concentrations are increased.“ Why not? The histogram of event sizes could not be fitted with a single Gaussian, but who is to say whether spontaneously fusing vesicles exactly follow a Gaussian distribution? The three distributions at 1q, 1.3q and 2q are not visible as peaks in the histogram at all; especially the fit of a distribution at 2q appears unreasonable. Finally, the suggestion that events with normalized amplitude at 1.3 q would result from multivesicular release of two vesicles with a fixed delay (118 ms) appears unreasonable; what mechanism would ensure that multivesicular release would happen always with this delay, especially as this is spontaneous release? I think the authors should remove this interpretation, which has consequences also for the interpretation of*
Figure 4
*and for the rates reported in*
Figure 5
*and possibly in*
Figure 6.

We have included alternative interpretations of our data in the text. Briefly, we cannot exclude that the larger events are the preferential fusion of slightly larger vesicles. However, we believe it is unlikely that the larger events are due to a change in the photophysical properties of pHluorin as earlier studies including our own did not detect significant difference in unitary fluorescence amplitudes as a function of extracellular Ca2+ (9, 25).

*3) The authors use the lack of a correlation between decay time and amplitude as an argument that “there is a fundamental difference in the kinetics of endocytosis and reacidification between vesicles that fuse spontaneously and those that fuse in response to stimulation.” But I would argue that this finding, together with the previous work of the authors,*
[25]
*is an argument against the interpretation of multivesicular release*.

We have added a note discussing this possibility in addition to our interpretation to the text.

*4) To follow up on the last statement above in point 2, to estimate the fusion frequencies (reported in*
Figure 5*) the authors “counted all events with amplitudes within the first quantal mean as a single event, and events with larger amplitudes as two events”. This is not appropriate, first because it is not clear whether larger events represent multiple events, and second because the distributions at 1q and 1.3q overlap. The authors should only use the number of events that they can detect as such, independently of amplitude. The same goes for*
Figure 6*, if the rates were calculated in the same way*.

We have clarified how we selected events in the text and have added both possible interpretations to the Figure (Figure 5).

*5) The correlation analysis in*
Figure 6
*has not been described in the statistics section of the Materials and Methods. Was this Pearson's correlation coefficient? The R^2 is quite small, but nevertheless the slope is positive under all circumstances. The authors should add a statistical test of the hypothesis R=0, and report whether there is a significant (but small) correlation. Finally, the authors should explain why all evoked fusion probabilities are integer multiples of 0.1 (panel C, D), or 0.05 (panel B). Why this difference between the conditions? Finally, given that fusion probabilities can be measured in only 10 (C, D), or 20 (B) discrete categories, would non-parametric correlation analysis be more appropriate?*

We agree, we have detailed the statistics used in the Materials and methods. Additionally we have used both a Pearson’s correlation coefficient as well as Spearman’s correlation coefficient. We now also report the p value of the R^2^ and r values for the Pearson’s and Spearman’s correlation values, respectively.

We now added the statistical test to compare the data to a hypothetical slope of zero. We found the slope of the regression line in 2 mM Ca^2+^ is significantly non-zero however because the R^2^ value is extremely low, it is difficult to interpret this finding as a strong correlation.

We have also explained the difference between fusion probabilities in the Materials and methods. Briefly, due to the low release probability in 2 mM extracellular Ca2+, we stimulated neurons twice as much with half the inter-stimulation interval to increase the number of events we detect.

Reviewer #3:

*1) Detection of single, spontaneous release events is a crucial technique for this paper. Though the amplitude distribution of the detected events looks convincing, it would also be useful to show how the distribution shifts when spontaneous release is blocked (i.e. with tetanus toxin incubation). Additionally, it is not clear how the noise events were detected. It would also be necessary to show how what the detection criteria discover when run with a negative amplitude (i.e. -2X the SD of the 17 points prior to the event)*.

We agree, however we opted to use extracellular medium with nominal Ca^2+^ to attenuate the rate of spontaneous release rather than attempt to abolish release using tetanus toxin. We chose this approach because previous work implicates the vesicular SNARE machinery may itself participate in and control the kinetics of endocytosis (Deak et al., 2004, Zhang et al., 2013) thus adding a layer of complexity to the experiments. In 0 mM extracellular Ca^2+^ and in the presence of folimycin, we only detected 29 genuine events (staircase patterns as in Figures 1 and 6) from 200 synapses over 10 minutes pooled from 4 coverslips. We found the amplitudes of false positive events, events that were detected using our criteria but where the amplitude returned to baseline despite folimycin presence, were smaller than what we detect in the case of events we identify as genuine spontaneous fusion. These small amplitudes of false positive events were similar to the amplitudes of the false negative events we identified when the detection criteria were run with a negative amplitude (i.e. -2X the SD of the 17 points prior to the event). This data are now shown in revised Figure 1L. Furthermore, in 2 mM Ca^2+^ we found when we ran our detection criteria with a negative amplitude we had a hit rate of 0.065 events per minute.

*2) For*
Figure 2*, the correlation between the spontaneous pHluorin events and spontaneous Ca2+ events was stated to be {plus minus} 1 s. While the limitations for time resolution in the dual color imaging are expected, since the vesicle release should directly cause the Ca2+ influx, these events should really only be considered if the Ca2+ influx occurs in a time range after the detected pHluorin event*.

We agree, however, in our imaging system we only acquire one time stamp for one frame of both green and red channel. Therefore, if a fusion event occurred during the acquisition of the red channel it is possible for the subsequent Ca2+ signal in the green channel to be detected first and thus be reported one “frame” earlier and thus >200 ms earlier. When we plot the temporal distribution of our events, we found that despite the ±1 s criterion, the majority of events occurred within ±1 frame (± 200 ms) of the red channel (please see the plot we present above). We have clarified this in the text.

*3)*
Figure 2*. What do the corresponding Ca2+ signals look like for the evoked SypHTomato events? Can they be more closely time-locked?*

Unfortunately we cannot image the corresponding Ca2+ signal for the evoked fusion events as leaving NMDARs unblocked will generate recurrent activity in our culture system since we must also exclude TTX from the bath solution (please see our response to summary comment #3 above).

*4)*
Figure 3*. Multivesicular release events recorded by electrophysiology are thought to occur essentially simultaneously. The 1.3q events seem likely to be events happening with a 118 ms delay. Also to this point, how well resolved are the single release sites? Could the multiple vesicle release events be closely neighboring synapses, which happen to release vesicles in close temporal proximity? Also*, *are the multivesicular spontaneous events Ca2+ dependent? How do they look in the presence of Cd2+?*

Our estimates of evoked and spontaneous fusion rates suggest that in the majority of boutons we detect release from a single release site. We agree that the role of voltage-gated Ca+ channels in driving these putative multivesicular events could be interesting to examine. However, we would like to perform these experiments using specific toxins rather than Cd2+. Ongoing work in our laboratory suggests that these events are unaltered in Cd2+, however, our data on this topic is too preliminary to report at this time.

*5)*
Figure 5*. The frequency of spontaneous release events clearly increases in 8 mM Ca2+ in the presence of folimycin. The authors explain this as recruitment of very rapidly retrieved vesicles. However, does the frequency of spontaneous events also increase in folimycin with 2 mM Ca2+? From the Figure, it doesn't appear to, but it would be interesting to see*.

The frequency of events in 2 mM Ca2+, if any, show a slight decrease in folimycin.

*6)*
Figure 6*. This is an important figure for the authors' interpretation of the separate pools of vesicles (spontaneous and evoked) released with distinct properties at single synapses. However, a few issues could cloud the interpretation from the lack of correlation between evoked fusion probability and spontaneous release frequency. First, for an event to be considered evoked with a 1 s lag time from the stimulus seems too long even for asynchronous release with a single action potential. Second, there is again the issue of spatial resolution. Are these actually single release sites? Finally, the release probability (the calculation of which should be clearly stated in the text) seems very high for these synapses in 4 mM Ca2+. This could argue for the multiple release site measurement*.

Our evoked release probability estimates at 2 mM Ca2+ (∼ 0.15) are in line with earlier measurements in hippocampal synapses. Therefore, we believe in a majority of cases we detect release from single release sites. It is important to note that hippocampal synapses do not show significant asynchronous release under physiological circumstances.

*7) Figure S1. The increase in signal for the PSD-95-GCaMP5K is difficult to interpret, because it could be due to either the increased extracellular Ca2+ or due to the increased multivesicular release events. This could also be interesting to see in the presence of Cd2+*.

We agree there could be multiple reasons for the increase in the Ca2+ signal. At this time, we prefer not to expand on this observation in depth particularly due to the fact that these signals are recorded in the absence of extracellular Mg2+. We are currently working on another study to examine the properties of postsynaptic Ca2+ signals triggered by spontaneous fusion events under more physiological circumstances.

[Editors’ note: further revisions were requested before acceptance, as shown below.]

1) Regarding detection of events, the authors added improved analysis. One reviewer would like to see addition of “false positive” events either from negative implement the detection (inversed) or from 0 Ca2+ to the histograms for amplitude and decay times. In addition, it would be helpful to subtract weighted average decay of false positives from weighted average decay of spontaneous events to obtain “net events”

We have included the false positive events in the decay times histogram (new Figure 1 panel D). Because we have already included the amplitudes in the cumulative probability histogram (Figure 1 panel L) we did not include the bars to the histograms. We have calculated a weighted average for the decay times and there was no change in decay time due to the relatively small contribution of false positive hits. We now refer to this data in the Results section (as well as in the revised Figure Legend for Figure 1).

*2) Coincidence of pre- and postsynaptic events: The detection of postsynaptic responses at the -1 frame will still be confusing to the reader, and additional clarification in the text is required. Given that the distribution of events puts most of them between 0 to 5 frames after stimulation, this is the time window that the authors should use for analysis. They will lose very few events and it will be more straightforward as it makes little sense that the postsynaptic signal should occur prior to the presynaptic signal. If the -1 frame events are to be included, the explanation for why these events would appear to occur before the stimulation, as included in the rebuttal letter must also be in the text*.

*Unfortunately, the postsynaptic signals coupled to the evoked Syp-Tomato signals are missing, and this is still an obvious gap in the paper. As the authors explain in the rebuttal letter, this is due to an understandable technical issues with field stimulation and network activity, although that the field stimulation itself causes a signal is strange when NMDARs are not blocked by Mg2+. Again, the authors should address this issue in the text. Pointing out to the reader that the recording the postsynaptic events with stimulation is technically difficult does not take anything away from the findings, and will also preempt questions from a reader who sees the comparison of the postsynaptic signals generated by evoked or spontaneous release as the obvious next step*.

We agree and we have included these arguments in the text.

*3) Release probability vs spontaneous release: There is only a 50% increase in the average spontaneous fusion rate with Ca2+ increase to 8 mM, while average release probability increases 3-fold. This figure could simply suggest that the release frequency of minis is much, much less sensitive to calcium than evoked release probability. This should be discussed by the authors*.

We agree. We now refer to this observation in the Discussion section.

*4) Sensitivity to detect individual events/multiquantal release: Overall, the single site, multiquantal spontaneous release claim is still not convincing (or at least multiple release site cannot be ruled out convincingly). I do not understand how the*
[31]
*citation excludes the possibility that the authors are not detecting coincident release from neighboring synapses and not the multi-quantal events they report-particularly as the release probability they claim will result in individual sites in 2mM Ca2+ and this will be less likely occur as Ca2+ increases. Additionally, aside from the increase in release probability with increased calcium, other factors that contribute to ability to differentiate individual synapses include the density of synapse and the amount of binning*.

*The discrepancy between mEPSC frequency increase in electrophysiological recordings and optical recordings of spontaneous events (though small) could suggest the events reported as multiquantal are, in fact, neighboring synapses. It would be worth testing when the portion of the population of events with 1.3 q and 2 q amplitudes are treated as individual events from neighboring synapses, does the frequency increase of the optical recordings better match the electrophysiological ones*.

We now refer to this issue more specifically with the following section inserted into the Discussion section.

“As noted above, at our resolution we cannot exclude the possibility that the observed increase in amplitude could be due to vesicle fusion from multiple release sites within a region of interest, especially if elevated Ca2+ concentrations can activate adjacent release sites resulting in apparent multivesicular release albeit originating from two neighboring release sites. The discrepancy between the increases in mEPSC frequency we detect in electrophysiological recordings versus optical recordings of spontaneous events could indicate that some of the multivesicular events are occurring at neighboring synapses activated at elevated Ca2+.”